# Navigating Conflicting Views: Harnessing Trust for Learning

**Jueqing Lu** [1]  **Wray Buntine** [2]  **Yuanyuan Qi** [1]  **Joanna Dipnall** [3]  **Belinda Gabbe** [3]  **Lan Du** [1]

## Abstract

Resolving conflicts is critical for improving the reliability of multi-view classification. While prior work focuses on learning consistent and informative representations across views, it often assumes perfect alignment and equal importance of all views, an assumption rarely met in real-world scenarios, as some views may express distinct information. To address this, we develop a computational trust-based discounting method that enhances the Evidential Multi-view framework by accounting for the instance-wise reliability of each view through a probability-sensitive trust mechanism. We evaluate our method on six real-world datasets using Top-1 Accuracy, Fleiss' Kappa, and a new metric, Multi-View Agreement with Ground Truth, to assess prediction reliability. We also assess the effectiveness of uncertainty in indicating prediction correctness via AUROC. Additionally, we test the scalability of our method through end-to-end training on a large-scale dataset. The experimental results show that computational trust can effectively resolve conflicts, paving the way for more reliable multi-view classification models in real-world applications. Codes available at: https://github.com/OverfitFlow/Trust4Conflict

## 1. Introduction

Multi-View Classification (MVC) plays a critical role in deep learning by greatly enhancing the ability to make accurate decisions through integrating multi-source information. Its effectiveness has been verified with the successful application in many domains such as autonomous driving (Yurtsever et al., 2020) and AI-assisted medical diagnostic systems (Kang et al., 2020). Most of the existing studies

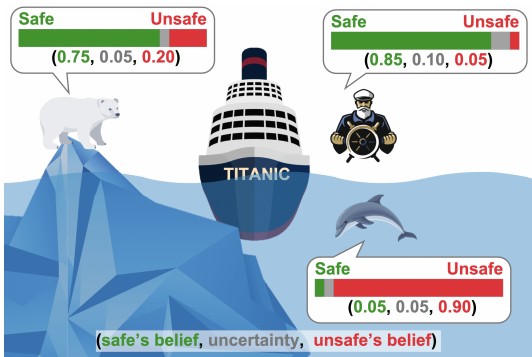

Figure 1. Example of conflicting multi-view opinions. The Titanic's route is safe in Captain's and Polar Bear's View, while unsafe in Dolphin's view.

on MVC rely on the assumption that data from different views consistently provide reliable information about the ground truth (Liang et al., 2024; Zhang et al., 2023a; Xu et al., 2024a). Nevertheless, this assumption may not always be valid in real-world scenarios. Substantial variations in the informativeness of data from different views can produce conflicting results, thereby undermining the reliability of the model's predictions.

A possible solution for resolving conflicts is to project data from different views into a shared latent space (Hardoon et al., 2004; Wang et al., 2015; Federici et al., 2020; Hjelm et al., 2019), and then draw a joint representation from the latent space for the classification task. This is achieved by integrating essential features via weighting schemes, such as attention mechanisms (Zheng et al., 2021) and weighted fusion (Atrey et al., 2010; Zhang et al., 2019). These methods typically assign higher weights to more informative views or features, thus reducing the impact of potential conflicting information. Although these methods have achieved promising results in MVC, their focus on the joint representation can be a limitation. Solely relying on the joint representation hinders the capacity to thoroughly grasp information provided by different views. In contexts such as ocean navigation, characterized by observations sources from various views (e.g., the perspectives of the captain, dolphin and Polar Bear when observing an iceberg as shown in Figure 1), it is crucial to thoroughly analyze and comprehend each

[1]Department of Data Science & AI, Monash University [2]College of Engineering and Computer Science, VinUniversity [3]School of Public Health and Preventive Medicine, Monash University. Correspondence to: Lan Du <Lan.Du@monash.edu>.

*Proceedings of the 42nd International Conference on Machine Learning*, Vancouver, Canada. PMLR 267, 2025. Copyright 2025 by the author(s).

view before making the decision to cross and face or detour, as different views provide unique and complementary information.

Existing approaches to resolve conflicts build neural networks to generate view-specific predictions and then combine view-specific predictions together. As a prime example, the Evidential Multi-view framework (Han et al., 2021) is emerging as a promising approach, offering a reliable means for the final fusion stage. Within this framework, evidence acts as a metric of endorsement for the associated predicted label, and the evidence is collected through view-specific neural networks. Subsequently, evidence from diverse viewpoints is fused, considering their respective epistemic uncertainties. However, there may exist cases where the view-specific information is not well aligned with the ground truth, resulting in misleading predictions with high confidence (low uncertainty). For example, as shown in Figure 1, while the dolphin can clearly observe the massive structure hidden beneath the water's surface, the captain may only see the tip of the iceberg.

In this work, we take a significant step further: leveraging the Evidential Multi-view framework, we propose a new computational trust based opinion fusion method to resolve potential conflicts in MVC. Specifically, the computational trust is modeled through an evidence network that operates on a view-specific and instance-wise basis. Drawing upon the principle of trust discounting in subjective logic, it evaluates the reliability of view-specific predictions generated by existing Evidential frameworks, such as Evidential Deep Learning (EDL) (Sensoy et al., 2018). Within the proposed method, each view-specific evidence is transformed into a degree of trust using the Binomial opinion theory (Jøsang, 2018). These degrees of trust are then utilized to establish uncertainty and a trust-aware opinion, ultimately facilitating the generation of reliable predictions. In summary, the contributions of this paper include:

1. We present a novel learnable trust-discounting mechanism to extend the widely-used Evidential MVC framework, enhancing its conflict resolution capabilities. Drawing from the Binomial opinion theory within subjective logic, it operates on a view-specific and instance-wise basis, adeptly resolving conflicts among views through a probability-sensitive trust discounting rule;

2. We develop a stage-wise training strategy to optimize the parameters of the proposed mechanism, which works robustly on different datasets;

3. We conduct extensive experiments on six real-world datasets, showing that our method outperforms the existing Evidential MVC methods, particularly on the datasets exhibiting large discrepancy among view-specific predictions. In addition, our method can also enhance the consistency among opinions derived from different views.

## 2. Related Work

**Multi-View Classification** leverages multiple data sources, offering varied perspectives on the same object, to enhance the classification performance. Recent advancements in MVC have focused on generating noise-robust representations through cluster-based (Huang et al., 2023; Wen et al., 2023a; Zhang et al., 2023b), self-representation-based (Hou et al., 2020), and partially view-aligned (Wen et al., 2023b; Huang et al., 2020) methods, harnessing the expressive power of deep neural networks. However, noise-robust representations may not fully resolve conflicts in opinions for a given data instance, as conflicts may arise by discrepant information from distinct views, and the discrepancy cannot be eliminated by addressing noises. Our method addresses this limitation by introducing a separate evidence network that evaluates the reliability of view-specific predictions and adjusts the final predictions according to the degree of trust.

**Trusted Multi-View Classification** has emerged as a crucial area and a pivotal domain within Multi-View Learning. This research area aims to enhance the accuracy and dependability of classification models by integrating data from multiple views, guided by their prediction confidence and epistemic uncertainty. The seminal work, Trusted Multi-View Classification (TMC) (Han et al., 2021), introduced the fusion of different views from an opinion perspective using the Dempher-Shafer Combination rule. Building upon TMC, Han et al., 2022 extended the approach by incorporating the pseudo-view, a concatenation of all other views, resulting in improved performance. Subsequent studies by Liu et al., 2022 and Xu et al., 2024a explored alternative opinion fusion methods. Concurrent research efforts, such as those by Jung et al., 2022 and Jung et al., 2023, focus on multiview uncertainty estimation, enhancing the model's reliability. Recently, TEF (Liang et al., 2025) proposed evolutionary fusion to enhance pseudo-view quality in Trusted Multi-View Classification. Similar to the TMC, our method is also built upon the Evidential Neural Network (ENN), but with a novel Trust Discounting module integrated, which adjust the original evidence and opinions before the Dempher-Shafer Combination based on the reliability of evidence and opinion.

**Conflictive Multi-View Classification** argues that existing work primarily focusing on either learning joint aligned representations or better quantifying uncertainty overlook the problem of potential contradictory in the prediction space. Recognizing this gap, the pioneer work by Xu et al., 2024a highlighted this issue and introduced the Degree of Conflict loss. This loss quantifies the disparity between different

predictions in the prediction space while accounting for uncertainty, aiming to mitigate conflict-related challenges. However, this approach may inadvertently lead correct predictions to converge towards incorrect ones, potentially jeopardizing model stability. In the case, if most views are making incorrect predictions, the minority of correctly predicted views may be forced to align with the majority of incorrect ones. In contrast, our method can generate more accurate predictions with properly estimated uncertainty. As the trust discount module of our method is trained based on the correctness of the view-specific prediction and directly assess the reliability of it, instead of using other views's predictions which may provide incorrect optimization direction.

## 3. Trust Fusion Enhanced Evidential MVC

### 3.1. Preliminaries

Given training data $\mathcal{D} = \{\{\boldsymbol{x}_i^v\}_{v=1}^V, y_i\}_{i=1}^N$ where $N$ is the number of training data, each instance $\boldsymbol{x}_i$ has $V$ views, ground truth label $y_i$ and an one-hot encoded label $\mathbf{y}_i$ (i.e., for a $K$-class classification problem, $\mathbf{y}_{i,k}$ is 1 if $k$ is the index of ground truth label for $i$-th instance, otherwise it is 0). The task of MVC is to learn a function $f$ that maps $\{\boldsymbol{x}_i^v\}_{v=1}^V$ to $\mathbf{y}_i$.

The Evidential MVC framework applies Subjective Logic (SL) to the $K$-class classification problem by assigning belief masses to individual class labels and computing epistemic uncertainty for the generated belief masses. The formulation links the evidence collected from instance view-specific observation to the concentration parameter of the Dirichlet Distribution. Let $f_\theta^v(\cdot)$ denote the view-specific neural network for evidence generation, where the view-specific evidence for an instance is $\boldsymbol{e}^v = f_\theta^v(\boldsymbol{x}^v)$, the association between the evidence and the Dirichlet parameters is simply $\alpha_k = e_k + 1$ (Sensoy et al., 2018; Han et al., 2021) . The belief mass on class label $k$, denoted as $b_k$, and uncertainty $u$ are subject to the additive requirement, i.e., $u + \sum_{k=1}^K b_k = 1$. With respect to MVC, the view-specific belief mass $b_k^v$ and uncertainty $u^v$ can then be computed as

$$S^v = \sum_{k=1}^K \alpha_k^v, b_k^v = \frac{e_k^v}{S^v} = \frac{\alpha_k^v - 1}{S^v}, u^v = 1 - \sum_{k=1}^K b_k^v = \frac{K}{S^v} \tag{1}$$

To generate the final prediction, SL models the view-specific predictions as multinomial opinions, denoted as $\omega^v = [\boldsymbol{b}^v, u^v, \boldsymbol{a}^v]$, with $\boldsymbol{a}^v$ being the base rate (i.e., a prior probability distribution over classes, generally a discrete uniform distribution), and then combine them together with an appropriate belief fusion rules based on the context (Jøsang et al., 2013). The Belief Constraint Fusion (BCF) (Jøsang et al., 2013), an extension of Dempher-Shafer combination rule (Shafer, 1976), was first adopted by Han et al., 2021 in

trusted MVC. Other fusion rules, such as Aleatory Cumulative Belief Fusion (A-CBF) (Liu et al., 2022) and Averaging Belief Fusion (ABF) (Xu et al., 2024a) have also been explored. We choose to stay with BCF in our experiments due to its intuitive foundation (Jøsang et al., 2013; Jøsang, 2018) and the effectiveness demonstrated by Han et al., 2021; 2022.

The fusion rule, $\oplus$, of BCF, among two views, i.e., $\omega = \omega^1 \oplus \omega^2$, can be formulated as follows:

$$b_k = \frac{1}{1-C}(b_k^1 b_k^2 + b_k^1 u^2 + b_k^2 u^1), \quad u = \frac{1}{1-C} u^1 u^2 \tag{2}$$

where $C = \sum_{i \neq j} b_i^1 b_j^2$ is the normalization factor, and $b_k$ is the belief mass of label $k$ and $u$ is the uncertainty the fused opinion $\omega$. Since the order of combination does not affect the final result (Jøsang, 2018), applying Eq. 2 by sequentially combining the $V$ views in pairs, where the result of each combination is then combined with the next view, will derive the final fused opinion, which is as follows,

$$\omega = \omega^1 \oplus \omega^2 \oplus \cdots \omega^V \tag{3}$$

For the fused opinion $\omega$, we can derive the parameters of the Dirichlet $\alpha_k$ by reversing the computation of Eq. 1.

**Corollary 3.1.** *An alternative representation for BCF is based on combining the evidence* [1]*, from which the opinion* $\omega = [\boldsymbol{b}, u, \boldsymbol{a}]$ *can be derived:*

$$e_k = e_k^1 + e_k^2 + \frac{e_k^1 e_k^2}{K} \tag{4}$$

### 3.2. Conflict Resolving By Trust Fusion

We realize conflicts can happen when view-specific opinions express conflicting preferences, leading to ambiguity in the fused opinion, for example, two views' candidate labels has same confidence(belief), and subsequently draws potential inaccurate predictions. Based upon this, we define the conflict problem as follows:

**Definition 3.2** (Conflicts within Multi-view Classification). In a $K$-class multi-class classification problem involving a multi-view dataset, a classification conflict arises when multiple views that predict different classes. This conflict leads to ambiguity in aggregating these predictions, as it becomes challenging to determine a single, coherent classification result from those inconsistent predictions.

Although Belief Fusion has been verified effectively to fuse different opinions under SL, it still can generate unreliable fused opinions and lead to inaccurate predictions, for example, the Titanic navigation route case used in Figure 1. The data of different views' opinions have been recollected, and

---

[1]We provide the proof in Appendix C.2 and we implement BCF based on this equation due to its computational efficiency.

*Table 1.* Opinions from Different views and BCF Fused opinion

|  | Belief($b$) | | Uncertainty($u$) |
|---|---|---|---|
| View | Safe | Unsafe | |
| Captain(functional) | 0.85 | 0.05 | 0.10 |
| Dolphin(functional) | 0.05 | 0.90 | 0.05 |
| PolarBear(functional) | 0.75 | 0.20 | 0.05 |
| Fused (via BCF) | 0.68 | 0.31 | 0.01 |

shown in Table 1. Besides, we also compute the fused opinion generated through BCF by substituting the data of three (i.e., Captain, Dolphin and PolarBear) functional opinions into Eq. 2 and Eq. 3, and the fused opinion has also been appended to the Table 1.

From Table 1, we can see that compared to the "unsafe" option, the fused opinion assigns a higher belief mass to the "safe" option (0.68 vs. 0.31). As a result, the prediction will be "safe", which is factually incorrect, as indicated in Figure 1. We attribute this error to insufficient evidence being collected, resulting in less belief mass supporting the factually correct option, "unsafe," in the opinions of both Captain and PolarBear. Additionally, the fused opinion exhibits lower uncertainty (0.01) compared to the original views' opinions (0.1, 0.05 and 0.05), however, the uncertainty is expected to be higher than that of all views to reflect the struggle among different opinions in the presence of conflict.

We utilize the principle of Trust Fusion (TF) by Trust Discounting (TD) (Jøsang et al., 2015) to handle the incorrect prediction caused by conflicting opinions. The basic idea of TD is to discount evidence or opinion from an individual view as a function of trust on that view. It can be used to weigh the current view-specific opinion according to the degree of trust, thus guiding the fusion process to generate more reliable prediction. Here we present a Probability-sensitive Trust Discounting rule, as show in Eq. 5, and use it in an instance-wise manner in our experiments as follows,

**Definition 3.3** (Instance-wise Probability-Sensitive Trust Discounting). For each view of each individual instance, the trust-discounted opinion is defined as

$$\breve{\omega}_i^v = \ddot{\omega}_i^v \otimes \acute{\omega}_i^v = \begin{cases} \breve{\boldsymbol{b}}_i^v = \ddot{p}_{t,i}^v * \acute{\boldsymbol{b}}_i^v, \\ \breve{u}_i^v = 1 - \ddot{p}_{t,i}^v * \left( \sum_{k=1}^K \acute{b}_{k,i}^v \right). \end{cases} \quad (5)$$

where $i$, $v$ are the index for $v$-th view of $i$-th instance, $\otimes$ indicates the TD operator, $\breve{\omega}$ denotes the discounted opinion, and $\ddot{\omega}$, $\acute{\omega}$ denote referral opinion and functional opinion [2] (e.g., opinions in Table 1), respectively. The scalar probability $\ddot{p}_t$ denotes the Degree of Trust (DoT), representing how much we are confident with the opinion given by the view-specific evidential model. Given Eq. 5, we fuse the trust-discounted opinions from $V$ views of $i$-th instance with

---

[2]Definitions of different opinion can be found in Appendix A.

*Table 2.* Referral Opinions of Different views

|  | Belief($b$) | | Uncer- | DoT($\ddot{p}_t$) |
|---|---|---|---|---|
| View | Trust | Distrust | tainty($u$) | |
| Captain(referral) | 0.6 | 0.3 | 0.1 | 0.65 |
| Dolphin(referral) | 0.9 | 0.0 | 0.1 | 0.95 |
| PolarBear(referral) | 0.2 | 0.7 | 0.1 | 0.25 |

*Table 3.* Discounted Opinions from Different views and BCF Fused opinion

|  | Belief($b$) | | Uncertainty($u$) |
|---|---|---|---|
| View | Safe | Unsafe | |
| Captain(discounted) | 0.55 | 0.03 | 0.42 |
| Dolphin(discounted) | 0.04 | 0.86 | 0.10 |
| PolarBear(discounted) | 0.19 | 0.05 | 0.76 |
| Fused (BCF) | 0.22 | 0.70 | 0.08 |

BCF by:

$$\bar{\omega}_i = \breve{\omega}_i^1 \oplus \breve{\omega}_i^2 \oplus \cdots \oplus \breve{\omega}_i^V$$
$$= \left( \ddot{\omega}_i^1 \otimes \acute{\omega}_i^1 \right) \oplus \left( \ddot{\omega}_i^2 \otimes \acute{\omega}_i^2 \right) \oplus \cdots \oplus \left( \ddot{\omega}_i^V \otimes \acute{\omega}_i^V \right) \quad (6)$$

Note that 1) the referral opinion is different from the functional opinion shown in Table 1, which aims for assessing reliability of corresponding views' functional opinion, and 2) comparing with original Probability-Sensitive TD (Jøsang et al., 2012), our proposed instance-wise manner takes into consideration the opinions reliability of each instance, instead of global reliability of view only.

According to Jøsang et al., 2015, the probability $\ddot{p}_t$ can be computed by $\ddot{p}_t = \ddot{b}_t + \ddot{a}_t * \ddot{u}$ [3] with $\ddot{a}$ being the uniformly distributed base rate, i.e., $\ddot{a}_t = 1/2$ for each individual instance on each view. Assuming we have the referral opinions for each view's functional opinion in Table 1, and defined in the Table 2. By substituting trust scores $\ddot{p}_t$ with the data in Table 2 and functional beliefs $\acute{\boldsymbol{b}}$ with the data in Table 1 in Eq. 5 and Eq. 6, we effectively apply TD to original functional opinions. This process enabled us to compute the discounted opinions for each view as well as their fused opinion through BCF combination, which is shown as in Table 3.

We can see that with the intervention of TD, the BCF fused opinion now assigns more belief mass to "unsafe," which aligns with the factual label. Additionally, the uncertainty of the fused opinion is now 0.08, which is rational given that Captain's and PolarBear's opinions have high uncertainty. Therefore, the decision aligning with Dolphin's opinion, which has significantly lower uncertainty than the others, is reasonable.

**Corollary 3.4.** *Above Eq. 3.3 also corresponds to updating*

---

[3]We prove that $p_t = b_t + a_t * u$ is equivalent to $p_t = \alpha_2/(\alpha_1 + \alpha_2)$ with the assumption that base rate $a_t$ is uniformly distributed in Appendix C.1.

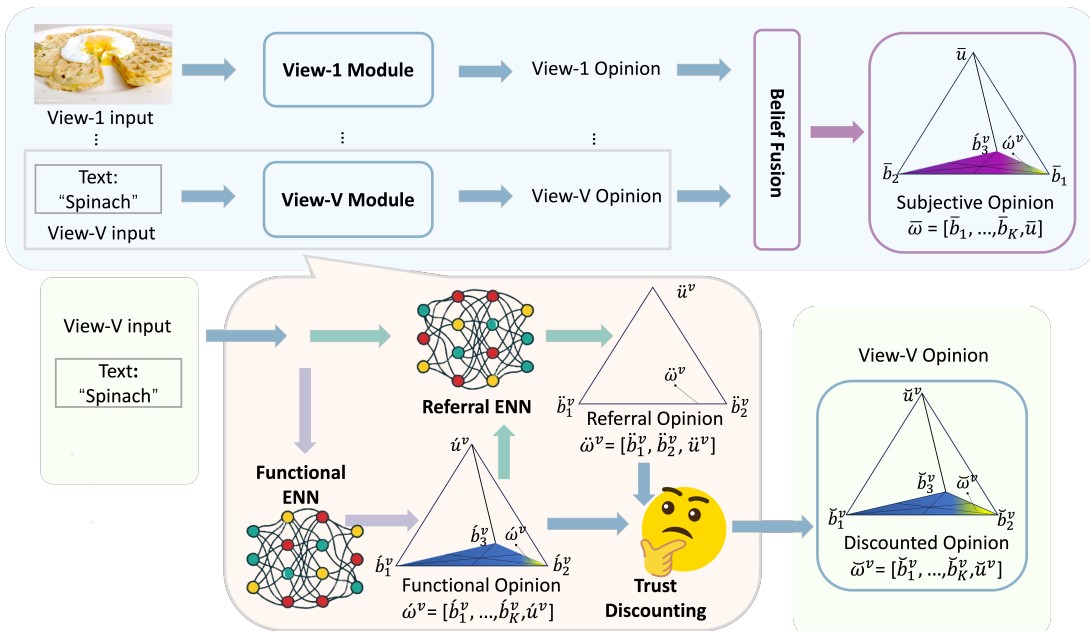

*Figure 2.* The TF Enhanced Evidential MVC Framework. The top half illustrates the overall pipeline of the Evidential MCV framework, while the bottom half zooms in to highlight the view-specific Trust Fusion.

the Dirichlet evidence by [4] :

$$\breve{e}_{k,i}^v = \frac{\ddot{p}_{t,i}^v \acute{u}_{t,i}^v}{1 - \ddot{p}_{t,i}^v + \ddot{p}_{t,i}^v \acute{u}_{t,i}^v} \acute{e}_{k,i}^v \tag{7}$$

The following propositions provide theoretical analysis of the proposed TD rule for achieving TF, and their detailed proof can be found in Appendix C.4.

**Proposition 3.5.** *Instance-wise Probability-Sensitive TD maximizes the belief mass of the Ground truth label after BCF, under the assumption that at least one view's prediction is correct.*

**Proposition 3.6.** *The combined opinion generated by proposed TF (TD+BCF) for conflicting views, will exhibit greater uncertainty than obtained through fusion with non-discounted functional opinions.*

### 3.3. Learning to Form Opinions

We depict the proposed TF (TD+BCF) along with entire Evidential MVC framework in Figure 2. The view-specific functional evidence is generated through an Evidential Neural Network (ENN), i.e., $\acute{e}_i^v = f_\theta^v(x_i^v)$, which is same as (Han et al., 2021). Similar to the functional evidence generation process, we construct another view-specific evidential network parameterized by $\ddot{\theta}$, for collecting referral evidence $\ddot{e}$, i.e., $\ddot{e}_i^v = f_{\ddot{\theta}}^v([x_i^v, \acute{b}_i^v])$ [5], where both feature representa-

tion $x_i^v$ and functional opinion $\acute{b}_i^v$ are used as inputs.

In terms of loss function, we follow Sensoy et al., 2018; Han et al., 2021; 2022; Xu et al., 2024a and optimize parameters of each view-specific evidential network. The loss term for $i$-th instance on $v$-th view is defined as follows,

$$L_i^v = \sum_{k=1}^K \mathbf{y}_{i,k}(\psi(S_i^v) - \psi(\alpha_{i,k}^v)) + \lambda_o D_{KL}[\mathrm{Dir}(\mathbf{p}_i^v|\tilde{\boldsymbol{\alpha}}_i^v)||\mathrm{Dir}(\mathbf{p}_i^v|\mathbf{1})] \tag{8}$$

where $\psi$ is the digamma function, $\lambda_o = min(1.0, o/10)$ is the annealing factor, and $o$ is the index of the current training epoch, $\tilde{\boldsymbol{\alpha}} = \mathbf{y} + (1-\mathbf{y}) \odot \boldsymbol{\alpha}$ is the Dirichlet parameters after removing misleading evidence from predicted distribution parameters $\boldsymbol{\alpha}$, and $\mathbf{p}$ is the projected probability, i.e., $\mathbf{p} = \boldsymbol{\alpha}/S$.

Note that, 1) the loss term above is directly linked with the distribution parameters that are generated through ENN parameterized by $\theta$, which will also be updated through back-propagation during training stage; 2) even though we omit the notation for distinguishing the distribution parameters that govern the variational transformation of referral and functional opinions, this loss term will still be applied to the referral and functional nets respectively; 3) the above equation will be also applied to the final fused opinion since its corresponding variational Dirichlet has parameter $\bar{\alpha}$ as well. We illustrate when and how to use the loss term in our

---

[4]We provide the proof in Appendix C.3.

[5]We use Bi-Linear layer instead of Dense/Linear Layer in our experiments.

---

**Algorithm 1** Algorithm For Training (simplified version)

---

**Input:** Multi-view dataset $\mathcal{D} = \{\{\mathbf{x}_i^v\}_{v=1}^V, y_i\}_{i=1}^N$.

**Initialize:** The parameters $\acute{\theta}, \ddot{\theta}$ of Functional and Referral ENNs, respectively.

**Stage-1 Warm-up Referral Network**

Obtain $\{\ddot{e}^v\}^V \leftarrow$ Referral ENNs outputs and $\{\ddot{\boldsymbol{\alpha}}^v\}^V$;

Update the parameters $\ddot{\theta}$ by Gradient Descent (GD) with loss of Eq. 10 for all $\{\ddot{\boldsymbol{\alpha}}^v\}^V$;

**Stage-2 Update Functional Network**

**/*Substage-2a*/**

Obtain $\{\acute{e}^v\}^V \leftarrow$ Functional ENNs outputs and $\{\acute{\boldsymbol{\alpha}}^v\}^V$;

Update the parameters $\acute{\theta}$ by GD with loss of Eq. 8 for all $\{\acute{\boldsymbol{\alpha}}^v\}^V$;

**/*Substage-2b*/**

Obtain $\{\ddot{e}^v\}^V \leftarrow$ Referral ENNs outputs and $\{\ddot{\boldsymbol{\alpha}}^v\}^V$;

Obtain $\{\acute{e}^v\}^V \leftarrow$ Functional ENNs outputs and $\{\acute{\boldsymbol{\alpha}}^v\}^V$;

Obtain $\ddot{\omega}^v$ and $\acute{\omega}^v$ by Eq. 1 with $\ddot{e}^v$ and $\acute{e}^v$ for all views;

Obtain BCF fused opinion $\bar{\omega}$ by Eq. 6 and $\bar{\boldsymbol{\alpha}}$ by Eq. 1;

Update the parameters $\acute{\theta}$ by GD with loss of Eq. 8 for $\bar{\boldsymbol{\alpha}}$;

**Stage-3 Adjust Referral Network**

By repeating Stage-2b and update $\ddot{\theta}$ instead of $\acute{\theta}$ only;

**Stage-4 Adjust Functional Network**

By repeating entire Stage-2;

**Output:** Functional and Referral networks parameters.

---

proposed stage-wise training algorithm ( Algorithm 1) [6].

We also adopt a warm-up stage for the referral nets since the randomly initialized parameters of them could introduce unreliable trust scores for discounting at early training stage. The loss term used at the warm-up stage is simply the left summation term of Eq. 8 with a different target label which is defined as

$$z_i^v = \begin{cases} 1 & \text{if } \hat{y}_i^v = y_i \\ 0 & \text{otherwise} \end{cases} \quad (9)$$

where $\hat{y}_i^v = \arg\max_k \acute{\boldsymbol{b}}$ which is predicted label of functional opinion, so the target label $z_i^v$ primarily indicates the correctness of such prediction. Following Müller et al., 2019, we apply label smoothing with smoothing factor $\eta = 0.9$ to the hard label. The association between one-hot encoded hard label $\mathbf{z}_i^v$ of target $z_i^v$ and smooth label is $\mathring{\mathbf{z}}_i^v = \mathbf{z}_i^v \odot \eta + (1 - \eta)/2$. since the smoothed label could provide training signals for neurons of both target and non-target labels, we omit the KL term here. The summation term, with Beta distribution parameters $\ddot{\boldsymbol{\alpha}}_i^v$ of referral opinion, changes to follows,

$$\sum_{j=1}^{2} \mathring{\mathbf{z}}_{ij}^v (\psi(\ddot{\boldsymbol{\alpha}}_{i1}^v + \ddot{\boldsymbol{\alpha}}_{i2}^v) - \psi(\ddot{\boldsymbol{\alpha}}_{ij}^v)) \quad (10)$$

---

[6]Due to space limitation, we provide a simplified version of training algorithm here for improving the readability and we direct readers to Appendix B for the detailed training algorithm.

# 4. Experiment

## 4.1. Experimental Setup

**Datasets.** Following previous work (Han et al., 2021; 2022; Jung et al., 2022; Xu et al., 2024a), we conducted experiments on six benchmark datasets: Handwritten[7], Caltech101 (Fei-Fei et al., 2004), PIE [8], Scene15 (Fei-Fei & Perona, 2005), HMDB (Kuehne et al., 2011) and CUB (Wah et al., 2011) with train-test split of 80% vs. 20%. A detailed description of these datasets is provided in the Appendix, we direct readers to the Appendix D.2 for further details regarding these datasets.

**Compared Methods.** We aim to resolve conflicts among predictions of different views, so we consider the methods that generate view-specific predictions which could have potential conflicts, and thus consider existing Evidential MVC baselines, TMC (Han et al., 2021), and the conflict resolution pioneering work ECML (Xu et al., 2024a). Recent work, TMNR (Xu et al., 2024b) applied Evidential MVC for noisy label learning, and CCML (Liu et al., 2024) derived consistent evidence among shared information by dynamically decoupling the consistent and complementary evidence [9]. Our method can also be extended to leverage the pseudo view, as demonstrated by its application to ETMC (Han et al., 2022), an extended version of TMC that incorporates pseudo views. We also compare with one multi-view uncertainty estimation baseline, MGP (Jung et al., 2022), in our experiments. We term our methods as TF and ETF where E indicates the pseudo-view is incorporated. All methods were run on a single 24GB RTX3090 card for fair comparison.

**Evaluation Metrics.** We evaluate MVC methods based on the reliability from prediction accuracy of fused opinion and the consistency among different views predictions. Similar to (Han et al., 2021; 2022; Jung et al., 2022; Xu et al., 2024a), we measure the prediction accuracy using Top-1 Classification Accuracy, which checks whether the final predicted label of fused opinion is same as ground truth. Regarding to the consistency among various views' predictions, we apply the Fleiss Kappa (Fleiss, 1971), which is a statistical measure for assessing the agreement between different raters, with scores closer to 1 indicating higher agreement among the different predictions. The intuition behind using this two metrics is a reliable prediction should not be accurate only but also from most agreements.

---

[7]https://archive.ics.uci.edu/ml/datasets/Multiple+Features

[8]http://www.cs.cmu.edu/afs/cs/project/PIE/MultiPie/Multi-Pie/Home.html

[9]We re-run the official implementation of ECML, TMNR, CCML with our data loader to ensure a fair comparison.

*Table 4.* Top-1 accuracy on test split. The best results are highlighted in **bold** and the second-best results are underlined.

| Method | Handwritten | Caltech101 | PIE | Scene15 | HMDB | CUB | AVG |
|---|---|---|---|---|---|---|---|
| MGP | 99.60±0.10 | 94.42±0.20 | 90.13±0.87 | 74.30±0.41 | 73.97±0.15 | 90.79±1.03 | 87.03 |
| ECML | 99.57±0.11 | 94.25±0.08 | 91.40±0.47 | 64.34±0.11 | 72.90±0.11 | 92.58±0.25 | 85.84 |
| TMNR | 99.72±0.08 | 94.31±0.09 | 89.34±0.59 | 74.14±0.13 | 73.46±0.15 | 92.25±0.38 | 87.21 |
| CCML | 99.00±0.00 | 94.64±0.10 | 93.09±0.36 | 73.97±0.15 | 72.59±0.42 | 93.83±0.41 | 87.91 |
| TMC | 99.63±0.13 | 94.30±0.13 | 87.43±0.90 | 73.99±0.19 | 73.30±0.18 | 92.50±0.37 | 86.60 |
| ETMC | 99.75±0.00 | 94.41±0.11 | 91.69±0.47 | 78.41±0.20 | 74.01±0.19 | 93.67±0.41 | 88.74 |
| TF (ours) | 99.68±0.11 | **95.26±0.10** | 93.31±0.40 | 77.83±0.32 | 74.35±0.09 | 93.33±0.75 | 88.96 |
| ETF (ours) | **99.98±0.07** | 95.07±0.08 | **94.63±0.34** | **82.01±0.17** | **75.55±0.15** | **94.08±0.38** | **90.22** |

*Table 5.* Fleiss' Kappa on test splits. The best results are highlighted in **bold** and the second-best results are underlined.

| Dataset | Handwritten | Caltech101 | PIE | Scene15 | HMDB | CUB | AVG |
|---|---|---|---|---|---|---|---|
| MGP | 0.59±0.05 | 0.94±0.00 | 0.21±0.01 | 0.33±0.00 | 0.51±0.00 | 0.43±0.07 | 0.50 |
| ECML | 0.42±0.05 | **0.95±0.00** | 0.40±0.01 | 0.26±0.00 | 0.53±0.01 | 0.44±0.07 | 0.50 |
| TMNR | 0.59±0.02 | 0.94±0.01 | 0.29±0.02 | 0.30±0.00 | 0.53±0.00 | 0.37±0.06 | 0.50 |
| CCML | 0.64±0.04 | 0.91±0.01 | 0.39±0.01 | 0.36±0.01 | 0.53±0.01 | 0.63±0.04 | 0.58 |
| TMC | 0.54±0.07 | 0.94±0.01 | 0.23±0.02 | 0.30±0.01 | 0.52±0.01 | 0.37±0.19 | 0.48 |
| ETMC | 0.66±0.01 | 0.84±0.00 | 0.28±0.04 | 0.37±0.00 | -0.15±0.04 | 0.45±0.10 | 0.41 |
| TF (ours) | 0.65±0.02 | **0.95±0.00** | 0.36±0.01 | 0.39±0.00 | 0.54±0.00 | 0.51±0.10 | 0.57 |
| ETF (ours) | **0.76±0.02** | **0.95±0.00** | **0.48±0.01** | **0.48±0.01** | **0.65±0.00** | **0.64±0.03** | **0.66** |

## 4.2. Experiment Results and Analysis

For each individual metric, mean and standard deviation from ten runs with ten different random seeds are reported. In all tables, the best-performing method is highlighted in bold, and the second-best method is underlined.

**Predictions Accuracy via Top-1 Accuracy.** Similar to (Han et al., 2021; 2022; Jung et al., 2022; Xu et al., 2024a), we first evaluated the model performance on the test split by Top-1 Classification Accuracy, as shown in Table 4. Building on the strengths of pseudo view, our method (ETF) consistently outperforms all baselines over six datasets. For example, on the PIE and Scene15 datasets, the use of referral trust boosts the accuracy of ETMC by 2.94% and 3.60%, respectively. Moreover, ETF surpasses the pioneering conflict resolving method ECML by a substantial margin of 3.23% on PIE, 9.66% on Scene15 and 2.65% on HMDB, highlighting better power of conflicts handling of our method. It is worth noting that Caltech101 inherently has lower level of conflicts, as corroborated by high accuracy and Fleiss' Kappa scores (Table 5) of all baselines.

When compared to well-established methods like TMC, MGP, and ECML without pseudo views, our method TF consistently demonstrates superior performance across all datasets. For example, our proposed trust discounting method enhance TMC's performance by 3.84% on Scene15 and 5.88% on PIE, while also achieving the highest Top-1 accuracy on other datasets. Notably, our method TF, even without incorporating pseudo views, exhibits comparable performance to ETMC with pseduo views. For instance, TF outperforms ETMC on three datasets (Caltech101, PIE, and HMDB) out of a total of six.

**Predictions Consistency via Fleiss' Kappa.** To further validate the effectiveness of our proposed method, we evaluate it with Fleiss' Kappa (Fleiss, 1971). our methods (ETF and TF) achieves the highest Fleiss' Kappa score on all six datasets (Handwritten, PIE, Scene15, HMDB and CUB). ETF enhances the robustness of ETMC with an improvement of approximately 13% on Caltech101. Moreover, it's essential to highlight that ETMC exhibits extremely poor agreement on HMDB with a negative value of -0.15. However, by applying our method, ETF significantly improves performance by an absolute value of 0.8. This underscores the relative robustness of our method across different datasets.

**Discussion on Consistency Improvement of Opinions from Different Views.** It is worth noting that applying TD solely on existing functional opinions cannot improve the consistency among different views, however, our methods show that the consistency of opinions from different views is significantly improved, as measured by Fleiss Kappa. We attribute this improvement to the incorporation of TD in the training stage. The functional opinion will be discounted accordingly by the referral opinion, and it thus receive larger magnitude of gradients from the loss term, e.g., Algorithm 1 stage 2b, due to interactions between different opinions, e.g., Eq.2. Therefore, the functional opinion will be enforced to align with the ground truth which leads to the improved consistency among different views' opinions.

## 4.3. Ablation Study

**Effectiveness of the TD module.** We conducted the ablation study to validate the effectiveness of TD module on

*Table 6.* Test Performance with or without the TD module.

| Method | Top-1 Acc(%) | Fleiss' Kappa |
|---|---|---|
| ETF(w/ TD) | 82.01±0.17 | 0.48±0.01 |
| ETF(w/o TD) | 81.06±0.16 | 0.46±0.01 |
| TF(w/ TD) | 77.83±0.32 | 0.39±0.00 |
| TF(w/o TD) | 76.82±0.33 | 0.37±0.01 |

*Table 7.* Test Performance with Different Smoothing Factors.

| Method | Top-1 Acc(%) | Fleiss' Kappa |
|---|---|---|
| ETF(0.9, reported) | 82.01±0.17 | 0.48±0.01 |
| ETF(1.0) | 82.07±0.12 | 0.48±0.01 |
| ETF(0.8) | 82.04±0.23 | 0.49±0.01 |
| ETF(0.7) | 82.07±0.10 | 0.48±0.01 |
| ETF(0.6) | 81.96±0.16 | 0.47±0.01 |

*Table 8.* Test Accuracy by using different views.

| view 1 | view 2 | view 3 | Top-1 Accuracy |
|---|---|---|---|
| ✓ | x | x | 57.16±0.22 |
| x | ✓ | x | 75.15±0.01 |
| x | x | ✓ | 62.97±0.45 |
| ✓ | ✓ | x | 78.70±0.00 |
| ✓ | x | ✓ | 68.21±0.01 |
| x | ✓ | ✓ | 80.21±0.00 |
| ✓ | ✓ | ✓ | 82.01±0.17 |

*Table 9.* Test Performance on Food101 via End2End training.

| Method | Top-1 Acc | Fleiss' Kappa |
|---|---|---|
| TMC | 92.35±0.34 | -0.0377±0.0130 |
| ETMC | 92.49±0.13 | 0.0252±0.0286 |
| ECML | 92.53±0.15 | -0.0207±0.0215 |
| CCML | 92.70±0.06 | -0.0342±0.0224 |
| TF (ours) | 92.79±0.15 | -0.0375±0.0255 |
| ETF (ours) | **93.09±0.02** | **0.0487±0.0228** |

Scene15. In the case without the TD module, the corresponding training stages in Algorithm 1 related to TD module will be disabled, for example, the warm-up stage and training stage 2b.

We can see from Table 6 that without the core module TD, the performance over four metrics drops, which indicates the effectiveness of our proposed TD module. It is also worth noting that, without TD, the model architecture is almost identical to TMC. However, both accuracy and Fleiss Kappa have improved, further demonstrating the effectiveness of our stage-wise training algorithm.

**Various Smoothing Factors.** We varied the smoothing factor used in the warm-up stage for ablation on Scene15. we set warm-up epoch equal to 1, which is same as the reported results in the main text. The equation we used for smoothing hard label is $\mathring{\mathbf{z}}_i^v = \mathbf{z}_i^v \odot \eta + (1 - \eta)/2$. With a larger smoothing factor, the smoothed label becomes meaningless, so we varied the factor from 0.6 to 1.0 by step size 0.1. According to Table 7, we can see that our method is relatively robust to different smoothing factors, and even gains performance improvement with adjusted smoothing factors on Scene15 Dataset, e.g., factor equals to 1.0, the smoothing factor we used in Table 4 (i.e., 0.9) is the empirical value suggested in the original paper, to avoid hyper-parameters over-tuning.

**Effectiveness of Leveraging Different Views.** We use the Scene15 dataset as an example and ablate the number of views to evaluate the performance of the trust discounting mechanism under varying numbers of views. From Table 8, we observe that the effectiveness of each individual view on classification varies significantly, as reflected in the test accuracy of individual views. However, our method consistently improves accuracy by effectively incorporating different views. The highest accuracy is achieved when all views are utilized together, which proves the effectiveness of our method.

### 4.4. End2End Training on Food101 Dataset

In order to further validate the effectiveness of our model, we use a larger dataset, Food101, which has both an image and text view. This is one dataset has the same number of class labels, 101, as Caltech101, and has more training (i.e., 61127), validation (i.e, 6845) and testing (i.e., 22716) instances. We train all methods using pre-trained Resnet50 and base-uncased Bert as image and text encoder, and we adopt AdamW Optimizer for fine-tuning parameters. All other settings, e.g., maximum number of epochs, are identical, and we run each method three times for reporting mean and standard deviation.

As indicated in Table 9, our method ETF consistently outperforms all other methods. Please note that TMNR is not included here as it requires pre-extracted feature vectors for computing similarity matrix, which works for noisy label learning and are kept frozen during training, but feature vectors are not able to be kept in this End2End training as the parameters of encoder will be updated.

## 5. Conclusion

In this paper, we introduced a theoretically-grounded approach for resolving conflicts in Multi-View Classification. This approach is built on top of the principle of the Trust Discounting in Subjective Logic, where the computational trust, aka referral trust, is represented as a Binomial opinion with a Beta probability density function.The functional trust is then discounted by the amount computed as a function of the degree of trust. We demonstrated through extensive experiments that the proposed trust discounting method not only benefits classification accuracy but also increases consistency among different views, providing a new reliable approach to handling conflicts in MVC.

## Acknowledgments

We would like to thank the anonymous reviewers for their valuable and helpful comments.

## Impact Statement

This research introduces a novel trust discounting mechanisms to address conflicts across multiple data views, and exhance the Evidential MVC framework. By training using the proposed stage-wise algorithm, our method improves classification accuracy and reliability in multiview data. These advancements have direct implications for applications in healthcare, autonomous systems, where reliability is critical. The findings provide a foundation for future work on adaptive trust mechanisms and conflict resolution in multiview classification system.

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

## A. The Definition of Opinions

A **Functional Opinion** expresses belief in a model's own ability to perform a certain task—such as a classification task. It reflects direct trust in the model's prediction. Let model $A$ A be evaluated for its ability to perform a function $f$ (e.g., classification). Then, a functional opinion is a subjective opinion represented as:

$$\acute{\omega} = [\acute{\mathbf{b}}, \acute{u}, \acute{\mathbf{a}}]$$

A **Referral Opinion**, in contrast, expresses belief in a model's ability to provide reliable referrals regarding another model's ability to perform a task. It reflects trust in the model's judgment, not in its own functional capability. Let model $B$ be asked to refer another model $A$ for the function $f$. A referral opinion captures our belief that model $B$ is reliable in making referrals about anther's (i.e., model $A$'s) ability to perform $f$, and is denoted as:

$$\ddot{\omega} = [\ddot{\mathbf{b}}, \ddot{u}, \ddot{\mathbf{a}}]$$

Regardless of whether the opinion is functional or referral, $\mathbf{b}$ is the belief mass vector, $\ddot{u}$ is uncertainty score, with $\mathbf{a}$ being the base rate (i.e., a prior probability distribution over classes, generally a discrete uniform distribution).

## B. Proposed Algorithm For Training and Testing

---

**Algorithm 2** Algorithm For Training

---

**Input:** Multi-view dataset: $\mathcal{D} = \{\{\mathbf{x}_i^v\}_{v=1}^V, y_i\}_{i=1}^N$.
**Initialize:** The parameters $\acute{\theta}$ of the Functional networks; initialize the parameters $\ddot{\theta}$ of the Referral networks.
**/\*Stage-1 Warm-up Referral Network\*/**
**for** minibatch **do**
    **for** $v = 1 : V$ **do**
        $\ddot{e}^v \leftarrow$ Referral Evidential network batch output;
        Obtain $\ddot{\alpha}^v \leftarrow \ddot{e}^v + 1$ ;
    **end for**Obtain overall loss by summing losses calculated by Eq. 10 of all $\{\ddot{\alpha}^v\}_{v=1}^V$;
    Update the parameters $\ddot{\theta}$ by gradient descent with the loss from above;
**end for/\*Stage-2 Update Functional Network\*/**
**for** minibatch **do**
    **/\*Substage-2a\*/**
    **for** $v = 1 : V$ **do**
        $\acute{e}^v \leftarrow$ Functional Evidential network batch output;
        Obtain $\acute{\alpha}^v \leftarrow \acute{e}^v + 1$ ;
    **end for**
    Obtain overall loss by summing losses calculated by Eq. 8 of all $\{\acute{\alpha}^v\}_{v=1}^V$;
    Update the parameters $\acute{\theta}$ by gradient descent with the loss from above;
    **/\*Substage-2b\*/**
    **for** $v = 1 : V$ **do**
        $\ddot{e}^v \leftarrow$ Referral Evidential network batch output;
        $\acute{e}^v \leftarrow$ Functional Evidential network batch output;
        Obtain $\ddot{\omega}^v$ and $\acute{\omega}^v$ by Eq. 1 with $\ddot{e}^v$ and $\acute{e}^v$, respectively ;
    **end for**
    Obtain joint opinion $\bar{\omega}$ by Eq. 6 and $\bar{\alpha}$ of this opinion by reversing Eq. 1;
    Obtain loss by Eq. 8 with $\bar{\alpha}$ and update the parameters $\acute{\theta}$ with gradient descent;
**end for/\*Stage-3 Adjust Referral Network\*/**
By repeating Stage-2b only and update $\ddot{\theta}$ instead of $\acute{\theta}$;
**/\*Stage-4 Adjust Functional Network\*/**
By repeating entire Stage-2;
**Output:** Functional and Referral networks parameters.

---

---

**Algorithm 3** Algorithm For Testing

---

**Requires:** The parameters $\acute{\theta}$ of the Functional networks; the parameters $\ddot{\theta}$ of the Referral networks.
**/\*Testing Phase\*/**
**for** minibatch **do**
    **for** $v = 1 : V$ **do**
        $\ddot{\mathbf{e}}^v \leftarrow$ Referral Evidential network batch output;
        $\acute{\mathbf{e}}^v \leftarrow$ Functional Evidential network batch output;
        Obtain $\ddot{\omega}^v$ and $\acute{\omega}^v$ by Eq. 1 with $\ddot{\mathbf{e}}^v$ and $\acute{\mathbf{e}}^v$, respectively ;
    **end for**
    Obtain joint opinion $\bar{\omega}$ by Eq. 6 and $\bar{\alpha}$ of this opinion by reversing Eq. 1;
    Obtain predicted labels of minibatch using $\arg\max$ over belief masses.
**end for**
**Output:** Predicted Labels and Opinions including fused opinion, functional opinions, referral opinions, discounted opinions for each instance of each view.

---

# C. Proofs And Derivations

## C.1. Calculation of Predictive Probability

According to Subjective Logic (SL) (Jøsang, 2018), the predictive probability $p_k$ for class $k$, can be calculated by

$$p_k = b_k + a_k * u \tag{11}$$

where $b_k$ is the belief mass for $k$-th label, $u$ is the predictive uncertainty or epistemic uncertainty (Sensoy et al., 2018). We usually assume the prior $a_k$ conforms to a uniform discrete distribution, i.e., $a_k = 1/K$, so the above equation is identical to

$$p_k = \frac{\alpha_k}{S} \tag{12}$$

where $\alpha_k$ is the Dirichlet concentration parameter for $k$-th label, and $S$ is the Dirichlet strength, i.e., $S = \sum_k \alpha_k$.

*Proof.*

$$
\begin{aligned}
p_k &= b_k + a_k * u \\
&= b_k + \frac{1}{K} * \frac{K}{S} \\
&= \frac{e_k}{S} + \frac{1}{S} \\
&= \frac{\alpha_k}{S}
\end{aligned}
$$

$\square$

Since Beta Distribution is 2-dimensional Dirichlet Distribution, above equations for calculating probabilities of multinomial opinions could also be applied to binomial opinions.

## C.2. Alternative Representation of Belief Constraint Fusion(BCF)

*Proof.* We the proof for Eq. 4 as follows,

$$
\begin{aligned}
e_k &= S * b_k \\
&= S \frac{1}{1-C}(b_k^1 b_k^2 + b_k^1 u^2 + b_k^2 u^1) \\
&= S \frac{1 - \sum_k b_k}{u^1 u^2}(b_k^1 b_k^2 + b_k^1 u^2 + b_k^2 u^1) \\
&= (S - S * \sum_k b_k) \frac{1}{u^1 u^2}(b_k^1 b_k^2 + b_k^1 u^2 + b_k^2 u^1) \\
&= (S - \sum_k e_k) \frac{1}{u^1 u^2}(b_k^1 b_k^2 + b_k^1 u^2 + b_k^2 u^1) \\
&= K \frac{1}{u^1 u^2}(b_k^1 b_k^2 + b_k^1 u^2 + b_k^2 u^1) \\
&= K \frac{1}{u^1 u^2}(\frac{e_k^1 e_k^2}{S^1 S^2} + \frac{e_k^1 u^2}{S^1} + \frac{e_k^2 u^1}{S^2}) \\
&= K(\frac{e_k^1 e_k^2}{K * K} + \frac{e_k^1 u^2}{K u^2} + \frac{e_k^2 u^1}{K u^1}) \\
&= \frac{e_k^1 e_k^2}{K} + e_k^1 + e_k^2
\end{aligned}
$$

$\square$

## C.3. Dirichlet Evidence Updating by Trust Discounting (TD)

As mentioned earlier, the TD in Definition 3.3 also corresponds to updating Dirichlet evidence using following equation,

$$
\check{e}_k = \frac{\ddot{p}_t \acute{u}}{1 - \ddot{p}_t + \ddot{p}_t \acute{u}} \acute{e}_k \tag{13}
$$

where $\ddot{p}_t$ is the probability representing trust degree and $\acute{u}$ is the uncertainty for functional opinion. $\acute{e}_k$ is Dirichlet evidence of functional opinion, and $\check{e}_k$ is Dirichlet evidence after discounting.

*Proof.*

$$
\begin{aligned}
\check{e}_k &= \check{\boldsymbol{b}}_k * \check{S} \\
&= \frac{\ddot{p}_t \acute{b_k} K}{\check{u}} \\
&= \frac{\ddot{p}_t \acute{b_k} K}{1 - \ddot{p}_t + \ddot{p}_t \acute{u}} \\
&= \frac{\ddot{p}_t}{1 - \ddot{p}_t + \ddot{p}_t \acute{u}} \frac{\acute{e}_k}{\acute{S}} K \\
&= \frac{\ddot{p}_t}{1 - \ddot{p}_t + \ddot{p}_t \acute{u}} \frac{K}{\acute{S}} \acute{e}_k \\
&= \frac{\ddot{p}_t \acute{u}}{1 - \ddot{p}_t + \ddot{p}_t \acute{u}} \acute{e}_k
\end{aligned}
$$

$\square$

## C.4. Detailed Proof of Propositions

*Proof.* Proof details of Proposition 3.5. Recall that scalar probability $\ddot{p}_t$ represents the degree of trust as mentioned before. The belief mass for $k$-th label of final fused opinion is as follows,

$$
\begin{aligned}
\bar{b}_k &= \frac{1}{1-\breve{C}}(\breve{b}_k^1 \breve{b}_k^2 + \breve{b}_k^1 \breve{u}^2 + \breve{b}_k^2 \breve{u}^1) \\
&= \frac{1}{1-\breve{C}}((\acute{b}_k^1 \ddot{p}_t^1)(\acute{b}_k^2 \ddot{p}_t^2) + \acute{b}_k^1 \ddot{p}_t^1 \breve{u}^2 + \acute{b}_k^2 \ddot{p}_t^2 \breve{u}^1)
\end{aligned}
$$

We use $g$ to denote the index of ground-truth label, and we have

$$
\bar{b}_g = \frac{1}{1-\breve{C}}((\acute{b}_g^1 \ddot{p}_t^1)(\acute{b}_g^2 \ddot{p}_t^2) + \acute{b}_g^1 \ddot{p}_t^1 \breve{u}^2 + \acute{b}_g^2 \ddot{p}_t^2 \breve{u}^1)
$$

The discounted opinion's uncertainty $\breve{u}$ is

$$
\begin{aligned}
\breve{u} &= 1 - \ddot{p}_t(\sum_k \acute{b}_k) \\
&= 1 - \ddot{p}_t(1 - \acute{u}) \\
&= 1 - \ddot{p}_t + \ddot{p}_t * \acute{u}
\end{aligned}
$$

In the warm-up training stage, the Eq. 10 is used to make sure $\ddot{p}_t \to 1$ (with hard targets for simplicity here) for those views' predictions are same as the ground truth label, and $\breve{u} \to 0$ for those views' predictions are incorrect. Therefore, $\breve{u} \to \acute{u}$ when $\acute{b}_g = max(\acute{\mathbf{b}})$, and $\breve{u} \to 1$ when $\acute{b}_g \neq max(\acute{\mathbf{b}})$.

Therefore, with the assumption that at least one-view's prediction is same the ground truth (i.e., correct label, let's say view 1's prediction is correct), we have

$$
\begin{aligned}
\bar{b}_g &= \frac{1}{1-\breve{C}}((\acute{b}_g^1 \ddot{p}_t^1)(\acute{b}_g^2 \ddot{p}_t^2) + \acute{b}_g^1 \ddot{p}_t^1 \breve{u}^2 + \acute{b}_g^2 \ddot{p}_t^2 \breve{u}^1) \\
&\geq \frac{1}{1-\breve{C}}((\acute{b}_k^1 \ddot{p}_t^1)(\acute{b}_k^2 \ddot{p}_t^2) + \acute{b}_k^1 \ddot{p}_t^1 \breve{u}^2 + \acute{b}_k^2 \ddot{p}_t^2 \breve{u}^1(\text{equality holds iif. } k=g)) \\
&= \frac{1}{1-\breve{C}}(\breve{b}_k^1 \breve{b}_k^2 + \breve{b}_k^1 \breve{u}^2 + \breve{b}_k^2 \breve{u}^1) = \bar{b}_k
\end{aligned}
$$

Besides the warm-up stage, in other training stages, such as training stage 3 in Alg.B, the $\ddot{p}_t$ will also be updated to maximize $\bar{b}_g$ based on the Eq. 8, i.e., $\bar{b}_g \geq \bar{b}_k$ (equality holds iif. $k = g$. Therfore, the referral opinion is learnt to maximize the belief mass of ground truth label of the final fused opinion as well.

$\square$

*Proof.* Proof details of Proposition 3.6. Let $\bar{u}$ and $\bar{u}'$ denote the uncertainty of BCF combined opinion with or without Trust

Discounting, respectively.

$$
\begin{aligned}
\bar{u} &= \frac{1}{\sum_{k=1}^{K}\left(\frac{\breve{b}_k^1 \breve{b}_k^2}{\breve{u}^1 \breve{u}^2} + \frac{\breve{b}_k^1}{\breve{u}^1} + \frac{\breve{b}_k^2}{\breve{u}^2}\right) + 1} \\
&= \frac{1}{\sum_{k=1}^{K}\left(\frac{\acute{b}_k^1 \ddot{p}_t^1 \acute{b}_k^2 \ddot{p}_t^2}{(\acute{u}^1 \ddot{p}_t^1 + 1 - \ddot{p}_t^1)(\acute{u}^2 \ddot{p}_t^1 + 1 - \ddot{p}_t^2)} + \frac{\acute{b}_k^1 \ddot{p}_t^1}{\acute{u}^1 \ddot{p}_t^1 + 1 - \ddot{p}_t^1} + \frac{\acute{b}_k^2 \ddot{p}_t^2}{\acute{u}^2 \ddot{p}_t^1 + 1 - \ddot{p}_t^2}\right) + 1} \\
&= \frac{1}{\sum_{k=1}^{K}\left(\frac{\acute{b}_k^1 \acute{b}_k^2}{\left(\frac{\acute{u}^1}{\ddot{p}_t^2} + \frac{1}{\ddot{p}_t^1 \ddot{p}_t^2} - \frac{1}{\ddot{p}_t^2}\right)\left(\frac{\acute{u}^2}{\ddot{p}_t^1} + \frac{1}{\ddot{p}_t^1 \ddot{p}_t^2} - \frac{1}{\ddot{p}_t^1}\right)} + \frac{\acute{b}_k^1}{\acute{u}^1 + \frac{1}{\ddot{p}_t^1} - 1} + \frac{\acute{b}_k^2}{\acute{u}^2 + \frac{1}{\ddot{p}_t^2} - 1}\right) + 1} \\
&= \frac{1}{\sum_{k=1}^{K}\left(\frac{\acute{b}_k^1 \acute{b}_k^2}{\left(\frac{\acute{u}^1}{\ddot{p}^2} + \frac{1 - \ddot{p}^1}{\ddot{p}^1 \ddot{p}^2}\right)\left(\frac{\acute{u}^2}{\ddot{p}^1} + \frac{1 - \ddot{p}^2}{\ddot{p}^1 \ddot{p}^2}\right)} + \frac{\acute{b}_k^1}{\acute{u}^1 + \frac{1}{\ddot{p}^1} - 1} + \frac{\acute{b}_k^2}{\acute{u}^2 + \frac{1}{\ddot{p}^2} - 1}\right) + 1} \\
&\geq \frac{1}{\sum_{k=1}^{K}\left(\frac{\acute{b}_k^1 \acute{b}_k^2}{\acute{u}^1 \acute{u}^2} + \frac{\acute{b}_k^1}{\acute{u}^1} + \frac{\acute{b}_k^2}{\acute{u}^2}\right) + 1} = \bar{u}'
\end{aligned}
$$

$\square$

### C.5. Loss Functions and Hyperparameters for Optimization

Recall that the probability density function (pdf) of the Dirichlet distribution, $\text{Dir}(\mathbf{p} \mid \boldsymbol{\alpha})$, is given by:

$$
\text{Dir}(\mathbf{p} \mid \boldsymbol{\alpha}) = \frac{1}{B(\boldsymbol{\alpha})} \prod_{i=1}^{K} p_i^{\alpha_i - 1}
$$

where:

- $\mathbf{p} = (p_1, p_2, \ldots, p_K)$ is a probability vector, such that $\sum_{k=1}^{K} p_k = 1$ and $p_k \geq 0$ for all $k$.

- $\boldsymbol{\alpha} = (\alpha_1, \alpha_2, \ldots, \alpha_K)$ is a vector of concentration parameters, with $\alpha_k > 0$.

- $B(\boldsymbol{\alpha})$ is the multivariate Beta function, defined as $B(\boldsymbol{\alpha}) = \frac{\prod_{k=1}^{K} \Gamma(\alpha_k)}{\Gamma\left(\sum_{k=1}^{K} \alpha_k\right)}$.

- $\Gamma(\cdot)$ is the Gamma function.

Recall that our loss function for Dirichlet Parameters $\boldsymbol{\alpha}$ is

$$
L_i^v = \sum_{k=1}^{K} \mathbf{y}_{i,k}(\psi(S_i^v) - \psi(\alpha_{i,k}^v)) + \lambda_o D_{KL}[\text{Dir}(\mathbf{p}_i^v \mid \tilde{\boldsymbol{\alpha}}_i^v) \| \text{Dir}(\mathbf{p}_i^v \mid \mathbf{1})]
$$

Specifically, the left summation term is derived from the Bayes risk for Cross-Entropy loss with a Dirichlet distribution, which is also dentoed as $L_{ace}$ in previous work (Han et al., 2021). We omit the index of view $v$ and instance $i$ for simplicity, so $L_{ace}$ is defined as follows,

$$
\begin{aligned}
L_{ace} &= \int \left[\sum_{k=1}^{K} -\mathbf{y}_k \log(p_k)\right] \frac{1}{B(\boldsymbol{\alpha})} \prod_{k=1}^{K} (p_k)^{\alpha_k - 1} d\mathbf{p} \\
&= \sum_{k=1}^{K} \mathbf{y}_k (\psi(S) - \psi(\alpha_k))
\end{aligned}
\tag{14}
$$

Where $\psi$ is the digamma function.

Recall that our referral network will generate the evidence for binomial opinion, and the evidence will be converted into parameters of Beta Distribution, i.e., $Beta(\alpha_0, \alpha_1)$ Subsequently, by replacing the Dirichlet Distribution with Beta Distribution, and the label $y_k$ in above equation with another label, we can have the $ace$ loss for Beta Distribution, as Eq. 10.

And the right term, KL divergence loss is

$$
\begin{aligned}
&\mathrm{D}_{KL}\left[\mathrm{Dir}(\mathbf{p} \mid \boldsymbol{\alpha}) \,\|\, \mathrm{Dir}(\mathbf{p} \mid \mathbf{1})\right] \\
&= \log\left(\frac{\Gamma\left(\sum_{k=1}^{K} \alpha_k\right)}{\Gamma(K)\prod_{k=1}^{K}\Gamma(\alpha_k)}\right) + \sum_{k=1}^{K}(\alpha_k - 1)\left[\psi(\alpha_k) - \psi\left(\sum_{j=1}^{K}\alpha_j\right)\right]
\end{aligned}
\tag{15}
$$

## D. Additional Details of The Experiment

### D.1. Hyper-parameters of Proposed Methods

The hyper-parameters for training TF and ETF has been shown in in Table 10. Concretely, "lr" is the learning rate for functional networks, "rlr" indicates the learning rate for referral networks. For the "lr", we follow ETMC (Han et al., 2022), and used same strategy to select learning rate for the functional nets. When tuning the learning rate for referral networks, we follow a basic principle of starting with a value less than or equal to the base learning rate, and then gradually decreasing the learning rate of referral network by a factor of three. For fair comparison, we used same learning rate for functional networks for evidence-based methods, except MGP (Jung et al., 2022), for which we followed their paper.

*Table 10.* TF and ETF hyper-parameters

| Hyper-parameter | Handwritten | Caltech101 | PIE | Scene15 | HMDB | CUB |
|---|---|---|---|---|---|---|
| lr | 3e-3 | 1e-4 | 3e-3 | 1e-2 | 3e-4 | 1e-3 |
| rlr | 3e-4 | 3e-5 | 1e-3 | 3e-3 | 1e-4 | 3e-4 |
| weight-decay | 1e-4 | 1e-4 | 1e-4 | 1e-4 | 1e-4 | 1e-4 |
| warm-up epochs | 1 | 1 | 1 | 1 | 1 | 1 |

The Adam optimizer (Kingma & Ba, 2015) is used for updating model parameters with beta coefficients = (0.9, 0.999) and epsilon = 1e-8.

### D.2. Summary of Dataset

*Table 11.* Summary of Datasets

| Dataset | Size | K | Dimensions | #Train | #Test |
|---|---|---|---|---|---|
| HandWritten | 2000 | 10 | 240/76/216/47/64/6 | 1600 | 400 |
| Caltech101 | 8677 | 101 | 4096/4096 | 6941 | 1736 |
| PIE | 680 | 68 | 484/256/279 | 544 | 136 |
| Scene15 | 4485 | 15 | 20/59/40 | 3588 | 897 |
| HMDB | 6718 | 51 | 1000/1000 | 5374 | 1344 |
| CUB | 600 | 10 | 1024/300 | 480 | 120 |

We provide the summary of the dataset in Table 11, we direct readers to (Han et al., 2021) for further details regarding these datasets. The datasets used in our experiments are 1) **Handwritten** dataset has 2000 samples of 10 classes. Each class is one of the digit 0 to 9 with samples evenly distributed (i.e., 200 samples per class). We use six descriptors to represent different views, and they are Pixel averages in $2 \times 3$ windows (Pix) feature with 240 dimensions, Fourier coefficients of the character shapes (FOU) with 76 dimensions, Profile correlations (FAC) features with 216 dimensions, Zernike moments (ZER) with 47 dimensions, Karhunen-Love coefficients (KAR) with 64 dimensions, and Morphological (MOR) features with 6 dimensions; 2) **Caltech101** dataset has 101 classes and 8677 images in total; We used the extracted features from DECAF (Donahue et al., 2014) and VGG19 (Simonyan & Zisserman, 2015). Both views have 4096 dimensions. 3) **PIE** dataset includes intensity (484 dimensions), Local binary patterns (LBP) (256 dimensions) and Gabor feature (279 dimensions) of 680 facial images, with 68 subjects; 4) **Scene15** dataset has 4485 images from 15 indoor and outdoor scene categories. There are 3

different views information, and they are GIST, Pyramid Histogram of Oriented Gradients (PHOG) and Local binary patterns (LBP) feature. These views are in 20, 59 and 40 dimensions respectively; 5) **HMDB** has 6718 samples of 51 categories of actions, which is consisted of Histogram of oriented gradients (HOG) feature and Motion Boundary Histograms (MBH) features as a 2-view dataset. Both views have 1000 dimensions; 6) **CUB** dataset has 200 different categories of birds and 11788 images in total. Same as (Han et al., 2021), we used first 10 categories in our experiment and GoogleNet (Szegedy et al., 2015) and doc2vec (Le & Mikolov, 2014) to extract the image features and text features to simulate a 2-view dataset. Image view and text view has 1024 and 300 dimensions respectively.

## E. Supplementary Insights and Additional Analysis

### E.1. Multi-View Agreement with Ground Truth (MVAGT)

The MVAGT (Multi-View Agreement with Ground Truth) is a novel evaluation metric designed specifically for multi-view classification problems with conflicting views. It assesses the model's performance on the test set by considering the ground truth labels, thus providing a more reliable and realistic measure of the model's ability to handle view disagreements. The rationality behind MVAGT lies in its alignment with real-world scenarios, where the majority agreement among multiple views is often considered more reasonable for the final decision. In the presence of view conflicts, a model that can make predictions consistent with the majority of views is deemed more trustworthy and reliable. By evaluating models using MVAGT, we can examine the reasonableness of the fused decision and assess the model's capability to handle view conflicts effectively. Mathematically, MVAGT calculates the accuracy of the model on the test set as follows:

$$\text{MVAGT} = \frac{1}{M} \sum_{i=1}^{M} \mathbb{1} \left( \sum_{v=1}^{V} \mathbb{1}((\hat{y}_i^v = y_i) > \frac{V}{2} \right) \tag{16}$$

where $M$ is the total number of test samples, $V$ is the number of views, $\hat{y}_i^v$ is the predicted label of the $i$-th sample from the $v$-th view, $y^i$ is the ground truth label of the $i$-th sample, and $\mathbb{1}(\cdot)$ is the indicator function that returns 1 if the condition is satisfied and 0 otherwise.

*Table 12.* MVAGT on test split. The best results are highlighted in **bold** and the second-best results are underlined.

| Dataset | Handwritten | Caltech101 | PIE | Scene15 | HMDB | CUB |
|---|---|---|---|---|---|---|
| MGP | 81.37±5.73 | 91.55±0.29 | 63.20±2.31 | 52.10±0.41 | 50.43±0.42 | 42.50±9.26 |
| ECML | 74.08±0.61 | 91.05±0.27 | 78.46±1.19 | 41.91±0.31 | 50.95±0.48 | 48.58±5.36 |
| TMNR | 86.80±1.03 | 90.92±0.18 | 65.15±3.68 | 51.86±0.61 | 50.48±0.47 | 36.58±6.42 |
| CCML | 86.78±1.42 | 88.97±1.09 | 81.91±1.40 | 55.23±0.84 | 51.34±0.91 | 63.67±2.61 |
| TMC | 81.58±6.57 | 90.27±0.38 | 51.54±3.00 | 51.42±0.46 | 50.37±0.45 | 43.25±14.8 |
| ETMC | 98.10±0.17 | 92.41±0.32 | 75.15±4.13 | 73.75±0.45 | 8.45±1.09 | 91.08±1.06 |
| TF (ours) | 88.97±0.61 | 92.01±0.22 | 80.59±0.75 | 60.41±0.52 | 52.47±0.35 | 54.33±7.54 |
| ETF (ours) | **98.53±0.08** | **94.47±0.12** | **90.37±0.40** | **79.18±0.38** | **71.43±0.32** | **91.17±0.67** |

### E.2. AUROC for Uncertainty.

The uncertainty score, as illustrated in Proposition 3.6, will be more accurate withou introducing biases, so it is essential to validate the increased uncertainty. Following the approach of prior work (Filos et al., 2019), we assess uncertainty to ensure a thorough evaluation. Specifically, we employed AUROC to measure the model's discriminate power in distinguishing incorrect predictions using uncertainty scores. As shown in Table 13, TF and ETF consistently demonstrate the best performance on five out of the six datasets, showcasing their robust generalizability. Despite a performance decrease on the CUB dataset, our method (ETF) still maintains the second-best result, outperforming other approaches, whether incorporating pseudo views or not. One possible reason for the decreased performance on CUB could be the unstable optimization caused by the limited number of training instances (e.g., 480), whereas other datasets, such as Scene15, contain significantly more instances (e.g., 3588).

### E.3. Ablation Study of Warm-up Epochs

In the proposed stage-wise training algorithm, we adopt a warm-up stage (i.e., training stage 1) for better initialization of referral networks. As random initialized parameters may not able to assess the reliability of corresponding functional

*Table 13.* AUROC of uncertainty scores for identifying incorrect predictions. The best results are highlighted in **bold** and the second-best results are underlined.

| Dataset | Handwritten | Caltech101 | PIE | Scene15 | HMDB | CUB |
|---|---|---|---|---|---|---|
| MGP | 99.29±0.30 | 87.62±0.90 | 88.43±0.67 | 63.92±1.96 | 82.87±0.60 | 58.20±11.4 |
| ECML | 79.05±5.62 | 86.31±0.50 | 87.51±0.49 | 60.50±0.25 | 81.63±0.15 | 57.30±8.50 |
| TMNR | 99.42±0.16 | 87.22±0.57 | 91.30±1.12 | 62.39±0.52 | 82.11±0.41 | 57.84±3.84 |
| CCML | 97.29±0.76 | 85.87±0.89 | 86.98±1.06 | 62.57±0.52 | 82.53±0.82 | 64.29±4.35 |
| TMC | 99.23±0.22 | 87.33±0.47 | 90.16±0.99 | 62.60±0.54 | 82.63±0.48 | 63.80±10.5 |
| ETMC | 99.30±0.19 | 88.35±0.63 | 93.02±1.40 | 66.49±0.44 | 85.42±0.34 | **72.56±8.11** |
| TF (ours) | 99.32±0.35 | **88.99±0.54** | **95.90±0.08** | 64.56±2.02 | 83.59±0.23 | 53.52±14.3 |
| ETF (ours) | **99.90±0.30** | 88.70±0.54 | 92.47±1.19 | **70.44±1.10** | **86.23±0.49** | 64.41±3.54 |

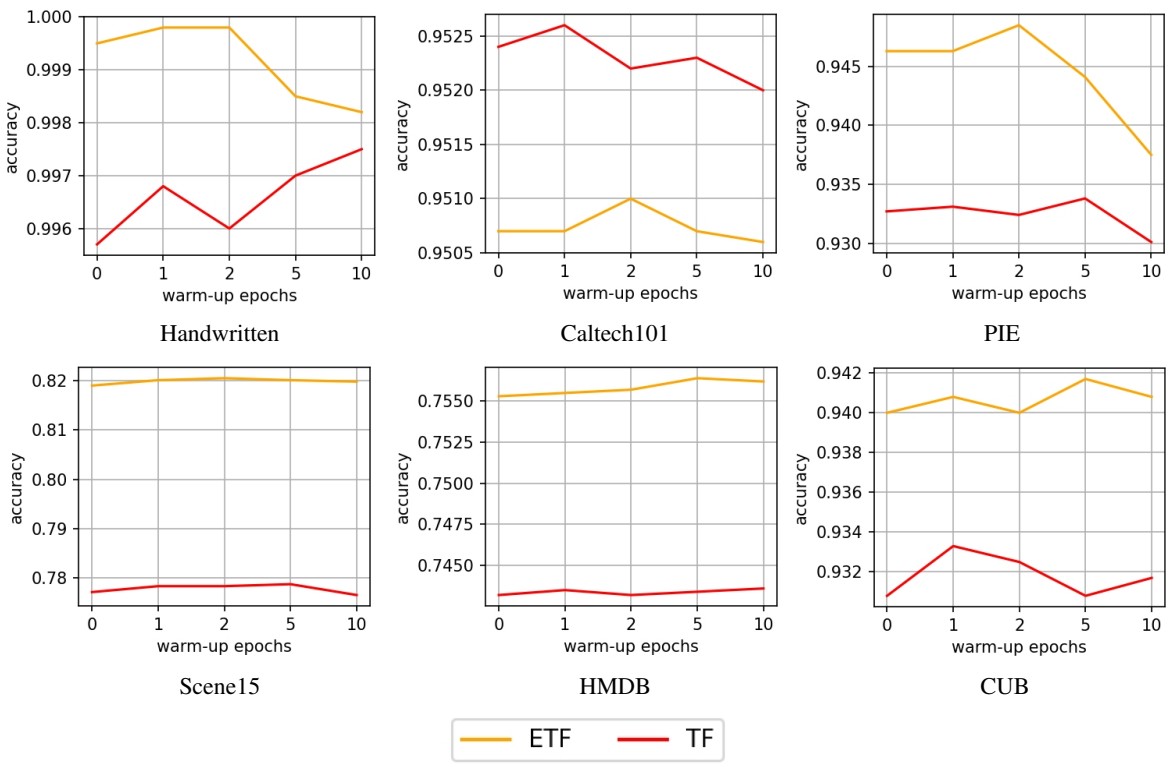

*Figure 3.* The effect of different warm-up epochs on testing accuracy.

opinions correctly. The key hyper-parameter of the warm-up stage, is the warm-up epochs. We ablate different values of this hyper-parameter and evaluate the effect of it on the performance of our method.

Specially, we used an empirical value, i.e., one single epochs, for all reported results in the experiment section. And here we provide more analysis with finely grain values, starting from 0 and increasing steadily, for example, to 2, 5, and 10, that is first random initializing the parameters of the referral networks and then not warm-up training or training with 2, 5, 10, and followed by each, finish the rest training stages. Please note that if this value is set to be 0, which means we disable the warm-up stage, and reported results with warm-up epoch 1 are also included, as shown in Figure 3.

From Figure 3, we can find that incorporating warm-up stage (warm-up epochs ≥ 1) can generally results in better accuracy. For some datasets (e.g. HMDB), increasing the number of warm-up epochs further improves accuracy compared to the results previously reported. This observation suggests that adjusting this value based on the specific dataset can lead to enhanced performance.

*Table 14.* View-Specific Pairwise Feature Similarity For Six Datasets

|  | View | Mean | Median | Min | Max |
|---|---|---|---|---|---|
| Handwritten | 1 | 0.6268 | 0.6329 | 0.1249 | 1.0000 |
|  | 2 | 0.8043 | 0.8095 | 0.4456 | 1.0000 |
|  | 3 | 0.8586 | 0.8592 | 0.6304 | 1.0000 |
|  | 4 | 0.7917 | 0.8038 | 0.2970 | 1.0000 |
|  | 5 | 0.9167 | 0.9168 | 0.8137 | 1.0000 |
|  | 6 | 0.7036 | 0.7964 | 0.0097 | 1.0000 |
|  | AVG | 0.7836 | 0.7889 | 0.5350 | 1.0000 |
| Caltech101 | 1 | 0.9684 | 0.9725 | 0.6968 | 1.0000 |
|  | 2 | 0.9748 | 0.9792 | 0.5175 | 1.0000 |
|  | AVG | 0.9716 | 0.9756 | 0.6263 | 1.0000 |
| PIE | 1 | 0.7518 | 0.7696 | 0.2842 | 0.9954 |
|  | 2 | 0.7173 | 0.7203 | 0.4939 | 0.8530 |
|  | 3 | 0.8613 | 0.8682 | 0.5598 | 0.9895 |
|  | AVG | 0.7768 | 0.7829 | 0.5471 | 0.9395 |
| Scene15 | 1 | 0.9038 | 0.9234 | 0.0538 | 1.0000 |
|  | 2 | 0.8689 | 0.8904 | 0.1185 | 1.0000 |
|  | 3 | 0.8133 | 0.8385 | 0.0072 | 1.0000 |
|  | AVG | 0.8620 | 0.8789 | 0.1170 | 1.0000 |
| HMDB | 1 | 0.9372 | 0.9375 | 0.9002 | 1.0000 |
|  | 2 | 0.9418 | 0.9418 | 0.8898 | 1.0000 |
|  | AVG | 0.9395 | 0.9397 | 0.8970 | 1.0000 |
| CUB | 1 | 0.4112 | 0.3952 | 0.1346 | 0.9577 |
|  | 2 | 0.9033 | 0.9128 | 0.5949 | 0.9972 |
|  | AVG | 0.6572 | 0.6494 | 0.4153 | 0.9674 |

## E.4. Instance Similarity of Vector Datasets

We also calculated the pair-wise cosine similarities and provided both the results and an analysis accordingly. Specifically, we considered to calculate the instance similarity using pair-wise cosine similarity. Please note the AVG view means calculating instance similarity on each view first, then averaging over all views.

Based on the Table above, we can see that for some datasets, like Handwritten and CUB, different views show different statistics indicating the similarity varies significantly in different views. However, for other datasets, like HMDB and Caltech101, the instance similarity among different views are pretty similar.

As we calculated the pairwise similarity using the feature vectors of instances, this similarity also reflects the semantic similarity. Consequently, similar statistics among different views suggest that their classification performance is likely to be comparable.

1) For similar views: If one view achieves high accuracy, the other is likely to perform similarly, resulting in both high accuracy and consistency. For example, this is observed in the Caltech101 dataset (refer to Top-1 Accuracy and Fleiss Kappa). If one view performs with low accuracy, the other tends to perform similarly, leading to fused predictions that are consistently low in accuracy across views. An example of this can be seen in the HMDB dataset.

2) For dissimilar views: If one view achieves high accuracy while the other produces low-accuracy predictions, this leads to higher conflicts. But the accuracy of the fused prediction depends on the specific fusion mechanism employed by the method. Examples of this scenario can be observed in the Handwritten and CUB datasets.

## E.5. Reduce Conflicts by Trust Fusion

We calculate the Conflict Ratio (CR) by normalizing the number of times that the $v$-th view prediction is different from $w$-th view, i.e., $\text{CR}(\hat{\boldsymbol{y}}^v, \hat{\boldsymbol{y}}^w) = \frac{1}{M} \sum_{i=1}^{M} \mathbb{1}(\hat{y}_i^v \neq \hat{y}_i^w)$, where $M$ is total number of test instances, $\hat{y}_i^w$ is the predicted label of $i$-th instance on $w$-th view, and $\mathbb{1}$ is the indicator function that returns 1 if the condition is satisfied and 0 otherwise. By applying Trust Discounting, both TMC's and ETMC's conflicts between different views are significant reduced. As an example, the CR on Scene15 is visualized by heatmap, shown in Figure E.5. The colors in the heatmap generated by our method are noticeably more blue (or less red) than those of the baselines, indicating that the conflict ratio has been reduced by our method.

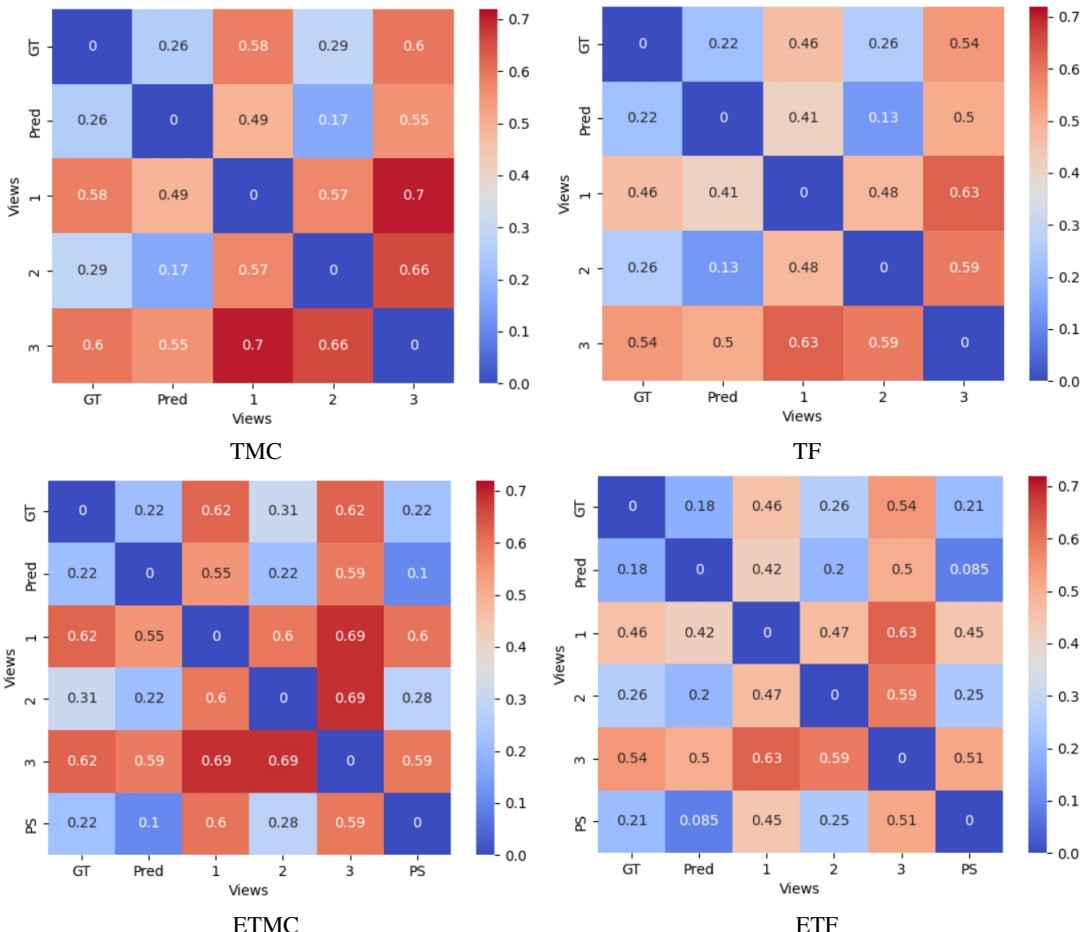

*Figure 4.* Conflict Ratio on Scene15, Four Methods TMC, TF, ETMC, ETF are compared. GT, Pred, 1, 2, 3 and PS are ground-truth, prediction, GIST, PHOG, LBP and pseudo view respectively.

### E.6. Explanation for the Decrease of AUROC for Uncertainty

We argue the decreased performance of AUROC on whether uncertainty can indicate the correctness of predicted label in caused by insufficient training instances. As shown in Table 11, there are less than 550 training instances on PIE and CUB datasets, where our methods, ETF and TF, have decreased performance, compared to ETMC and TMC, in which the only difference is the TD module.

Besides, we also investigate a particular testing instance of CUB dataset for the decreased performance on AUROC of uncertainty. As the error case displayed in Figure 5, ETF corrects the error prediction made by ETMC. However, even though the combined prediction is correct after applying trust discounting, the predictive uncertainty is still relatively high. If ETF corrects previously incorrect predictions but assigns them relatively high uncertainty scores (e.g., 0.4), it may lead to a decrease in the AUROC for predictive uncertainty. This is because AUROC evaluates the model's ability to discriminate between correct and incorrect predictions based on uncertainty scores. Correcting predictions while maintaining high uncertainty scores can make it more challenging for the model to distinguish between correct and incorrect predictions, resulting in a lower AUROC score, even though the accuracy improves.

### E.7. Simulating Conflicting Predictions with Noisy Instances

We plot the model performance for Evidential MVC methods with various level of noises introduced to inputs in Figure 6 and Figure 7, for methods incorporate pseudo views and not incorporate pseudo views respectively. Our methods consistently outperforms other methods like TMC and ECML.

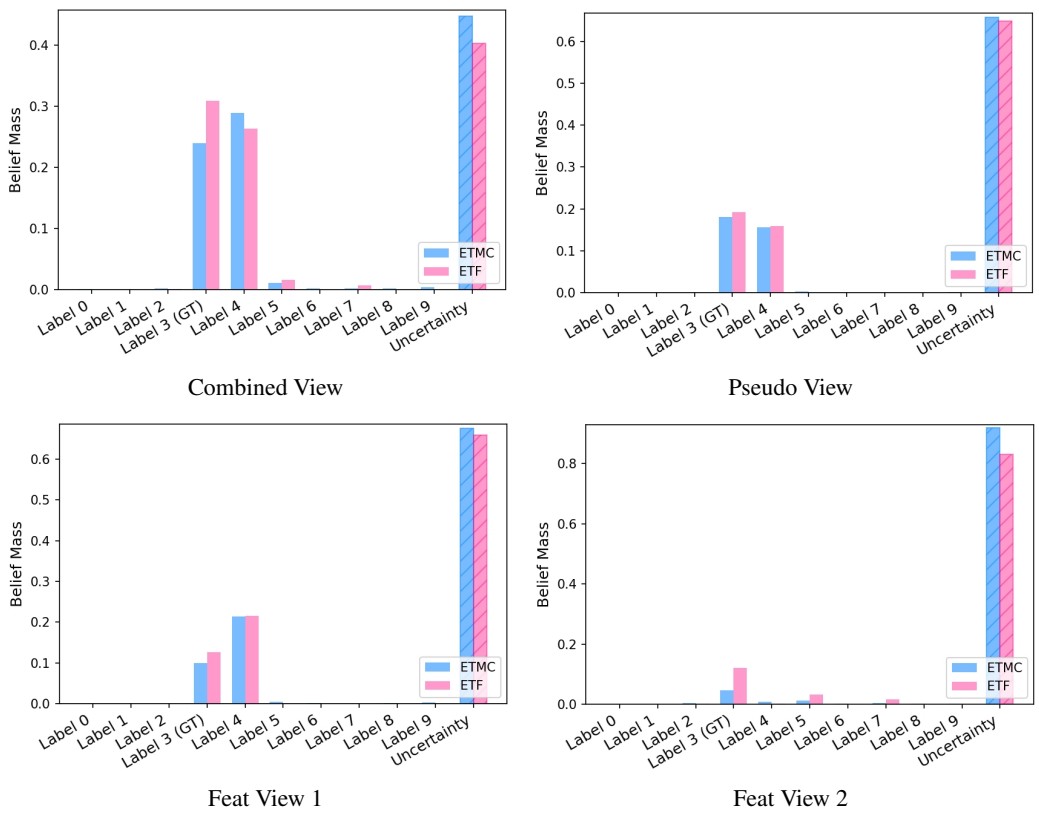

*Figure 5.* Bar chart for each label's belief mass and predictive uncertainty of one testing instance of CUB dataset. GT indicates the ground truth label of the selected instance.

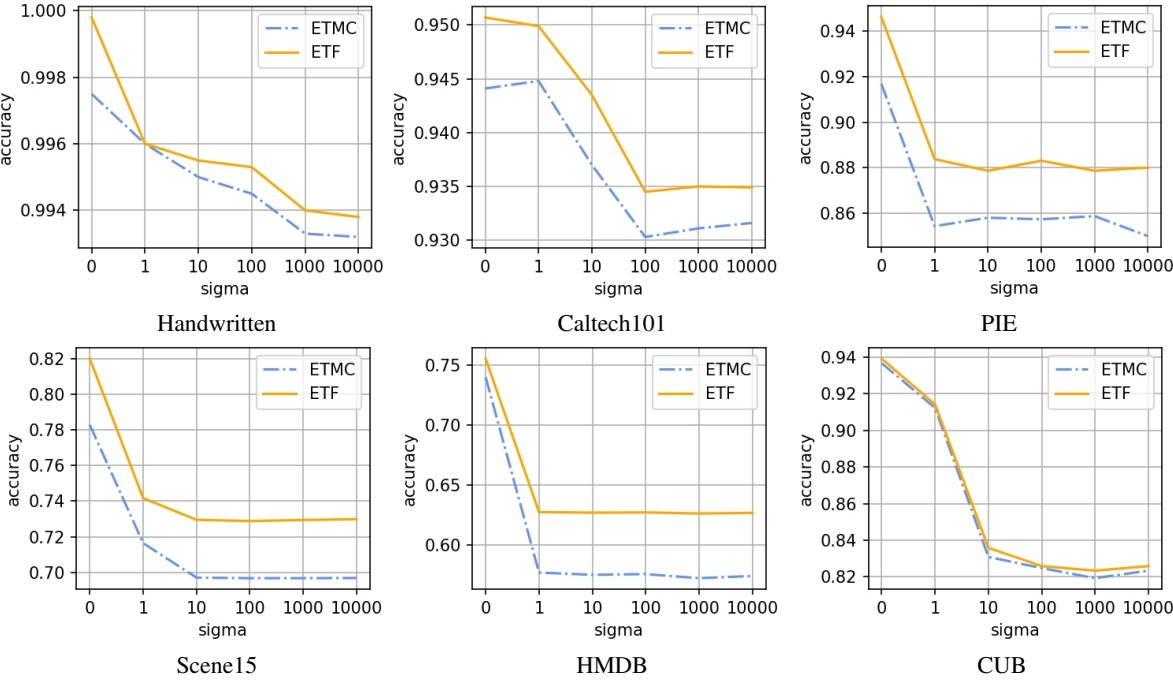

*Figure 6.* Performance of pseudo-view incorporated Evidential MVC methods on multi-view data with different levels of noise.

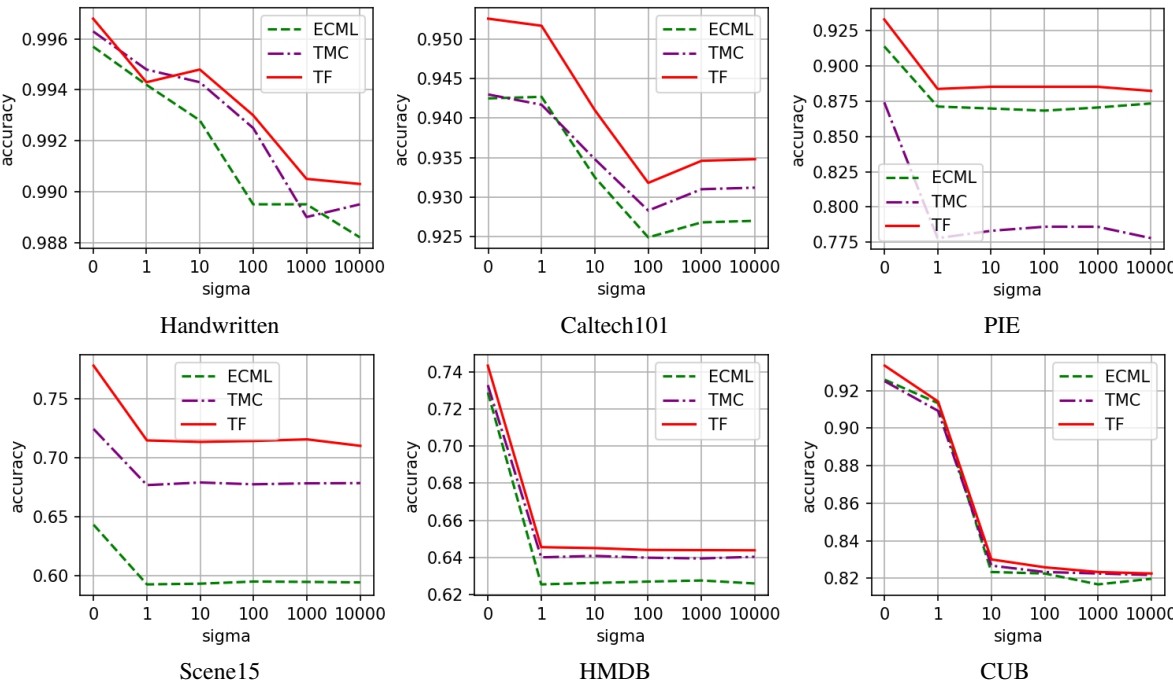

*Figure 7.* Performance of non pseudo-view incorporated Evidential MVC methods on multi-view data with different levels of noise.

## F. Technical Requirement and Execution

### F.1. Limitations

One possible limitation of our work is that the warm-up loss is not optimal solution, even though we explored the impact of different warm-up epochs and showed the effectiveness with using warm-up loss. Another possible limitation would be stage-wise training algorithm is time consuming, we leave it to future work for improving its efficiency.

### F.2. Execution Time

The proposed instance-wise approach does indeed introduce additional time complexity compared to the baselines, particularly compared to methods like TMC and ETMC that do not incorporate the TF Module but with same Belief Fusion method. However, our method does not rely on the dependencies between instances for computation. This allows us to perform batch-wise calculations during both training and testing, a practice widely adopted in most deep learning algorithms, which can enhance efficiency.

From another perspective, we can view the TF stage as an additional layer appended to the existing framework (e.g., TMC). Let $h$ be the input vector with dimension $d_h$ used for the classification task. For a $K$-class classification problem, we obtain a $K+1$-dimensional functional opinion (1 dimension for uncertainty). The weight matrix $W$ of the proposed BiLinear layer will have dimensions $d_h$ x $d_{K+1}$ x $d_2$, and the bias vector will have dimension $d_2$. The time complexity for matrix multiplication is O($d_h$ x $d_{K+1}$ x $d_2$) and the time complexity for bias addition is O($d_2$). Thus, the overall time complexity is O($d_h$ x $d_{K+1}$ x $d_2$). Given the dataset for a classification task, the additional layer exhibits linear time complexity with respect to only the hidden size. Since this hidden size is relatively small and compact to the classification dimension, we argue that the increase in time complexity is not substantial as shown in following tables. We report the training and testing time by averaging 10 times running as shown in Tables 15 - 20.

### F.3. Framework and Reproducibility

For experimental results to be reproducible, we will release our official implementation upon the paper's acceptance. Specifically, we used PyTorch (Paszke et al., 2019) version 1.13.0, built with CUDA 11.7, to implement our codes. The Python environment version is 3.8, and the operating system is Ubuntu 22.04.4. All Experiments are conducted on a single

*Table 15.* Handwritten

| Method | Train(Seconds) | Test(Seconds) |
|---|---|---|
| F-Avg | 22.88±0.30 | 0.040±0.09 |
| F-Mode | 26.26±0.36 | 0.041±0.09 |
| MGP | 452.31±1.43 | 0.428±0.10 |
| EMCL | 52.63±1.15 | 0.041±0.09 |
| TMC | 55.46±0.78 | 0.042±0.09 |
| TF | 183.51±1.81 | 0.043±0.09 |
| ETMC | 62.45±0.95 | 0.042±0.09 |
| ETF | 202.15±2.24 | 0.044±0.09 |

*Table 16.* Caltech101

| Method | Train(Seconds) | Test(Seconds) |
|---|---|---|
| F-Avg | 78.62±0.95 | 0.063±0.09 |
| F-Mode | 94.01±0.87 | 0.063±0.09 |
| MGP | 2439.60±7.35 | 3.428±0.13 |
| ECML | 152.99±5.96 | 0.064±0.10 |
| TMC | 114.77±1.89 | 0.066±0.10 |
| TF | 463.41±10.65 | 0.067±0.09 |
| ETMC | 153.64±1.690 | 0.066±0.09 |
| ETF | 543.99±24.88 | 0.067±0.010 |

*Table 17.* PIE

| Method | Train(Seconds) | Test(Seconds) |
|---|---|---|
| F-Avg | 4.94±0.26 | 0.033±0.09 |
| F-Mode | 6.06±0.27 | 0.034±0.09 |
| MGP | 123.63±2.38 | 0.374±0.11 |
| ECML | 12.92±1.50 | 0.035±0.09 |
| TMC | 11.39±0.31 | 0.035±0.09 |
| TF | 41.63±0.68 | 0.037±0.09 |
| ETMC | 10.36±0.37 | 0.036±0.09 |
| ETF | 50.39±0.71 | 0.037±0.09 |

*Table 18.* Scene15

| Method | Train(Seconds) | Test(Seconds) |
|---|---|---|
| F-Avg | 27.33±0.37 | 0.039±0.09 |
| F-Mode | 33.77±0.65 | 0.040±0.09 |
| MGP | 576.76±1.27 | 0.420±0.15 |
| ECML | 63.24±0.72 | 0.040±0.09 |
| TMC | 73.26±0.53 | 0.042±0.10 |
| TF | 229.05±2.86 | 0.042±0.09 |
| ETMC | 86.81±3.11 | 0.042±0.09 |
| ETF | 271.99±2.26 | 0.043±0.09 |

*Table 19.* HMDB

| Method | Train(Seconds) | Test(Seconds) |
|---|---|---|
| F-Avg | 38.26±0.65 | 0.045±0.09 |
| F-Mode | 48.86±0.64 | 0.048±0.09 |
| MGP | 654.42±1.35 | 0.971±0.13 |
| ECML | 82.32±1.17 | 0.047±0.09 |
| TMC | 74.62±0.65 | 0.047±0.09 |
| TF | 278.99±3.47 | 0.047±0.09 |
| ETMC | 99.54±0.93 | 0.046±0.09 |
| ETF | 365.94±8.12 | 0.047±0.09 |

*Table 20.* CUB

| Method | Train(Seconds) | Test(Seconds) |
|---|---|---|
| F-Avg | 3.57±0.29 | 0.033±0.09 |
| F-Mode | 4.48±0.29 | 0.033±0.09 |
| MGP | 136.74±0.76 | 0.239±0.10 |
| ECML | 8.17±0.28 | 0.036±0.09 |
| TMC | 7.66±0.30 | 0.034±0.09 |
| TF | 29.21±0.41 | 0.035±0.09 |
| ETMC | 13.98±0.38 | 0.035±0.09 |
| ETF | 37.57±0.56 | 0.036±0.09 |

Nvidia RTX 3090 GPU with 24GB of memory.

## G. More Discussions

### G.1. Comparison to ECML

The differences between ECML and our work can be summarized as follows,

1. Different Conflict Resolving Mechanism: our method uses a trust discounting module to modulate the trust on the functional opinions, while ECML uses a loss function to harmonize different views' functional opinions (this is already mentioned in the related work section).

2. Our method is built upon TMC and ETMC, so using the same belief fusion method, which is the Dempher-Shafer rule for combining different views from opinion perspective. However, ECML uses a different, evidence averaging based method to fused opinions.

3. To keep the effectiveness of the introduced TD module, we proposed stage-wise training algorithm, which is also different from ECML.

### G.2. Comparison to Original Trust Discounting

1. While the original subjective logic framework proposes a global trust-discounting mechanism, we extend it to operate in an instance-wise manner. This critical advancement enables adaptive handling of varying conflict patterns across

different samples, significantly enhancing the framework's flexibility.

2. We introduce a novel stage-wise training approach that is essential for keeping the effectiveness of our Trust-Discounting (TD) module. Based on our study, simply incorporating the TD module into existing training frameworks (e.g., ETMC's approach) would actually degrade performance. Our designed training strategy ensures stable optimization and reliable conflict handling.

### G.3. Potential Limitations

We analyze potential limitations from both global and local perspectives.

1. From a global perspective, the evidential multi-view classification framework relies on late fusion. This means that there is no early interaction between views during feature extraction. Although effective in many cases, this design might limit knowledge integration when different views provide complementary but partial information about the complete pattern.

2. From a local perspective, our Trust Discounting (TD) module has a potential limitation: it may still use referral opinions with high uncertainty. Since high uncertainty suggests lower reliability, this indicates that our current trust adjustment mechanism could benefit from more fine-grained handling of such cases. We acknowledge this as an area for future improvement.

