# OpenReview forum: "Navigating Conflicting Views: Harnessing Trust for Learning"
_ICML.cc/2025/Conference — ICML 2025 poster_

### Official Review · Reviewer_ngw9 · 2025-03-08

**Overall Recommendation:** 3

**Summary:**

This paper introduces a novel approach to resolving conflicting predictions in multi-view classification by integrating an instance-wise, probability-sensitive trust discounting mechanism within an evidential framework. The method computes a degree of trust for each view through a referral network, which is then used to discount the functional predictions on a per-instance basis before fusing them using BCF. A stage-wise training strategy is also proposed to alternately optimize the functional and referral networks, leading to improved prediction accuracy and consistency across various datasets, particularly in scenarios where different views provide contradictory information.

## update after rebuttal

The authors have addressed several of my concerns, and I appreciate the detailed discussion and experimental analyses provided in their rebuttal. I decided to maintain my rating.

**Claims And Evidence:**

Yes. I think the claims are supported.

**Essential References Not Discussed:**

No additional essential references appear to be missing. However, it is recommended that the authors discuss two newly accepted ICLR’25 papers related to Trusted Multi-View Classification to further contextualize and strengthen the contributions of this work.

**Experimental Designs Or Analyses:**

Q1. The benchmarks used are standard classification datasets that may not naturally exhibit strong conflicting views. Additional experiments on synthetic or real-world datasets specifically engineered to induce view conflicts could provide more direct validation of the method's core motivation.

Q2. The analysis primarily emphasizes classification accuracy and consistency, but further evaluation—such as direct measures of conflict (e.g., disagreement ratios between views) or error analysis on instances with high inter-view disagreement—would strengthen the conclusions regarding conflict resolution.

**Methods And Evaluation Criteria:**

The proposed methods are well-motivated and generally appropriate for addressing conflicts in multi-view classification. The trust discounting mechanism, combined with the stage-wise training strategy, effectively integrates into the evidential framework, enhancing both prediction accuracy and consistency. However, the evaluation primarily relies on standard classification benchmarks (e.g., Handwritten, Caltech101, PIE, etc.) and metrics like Top-1 Accuracy and Fleiss’ Kappa. While these are valid, they might not fully capture the nuanced challenge of handling view conflicts. Additional evaluation criteria—such as direct conflict quantification metrics or experiments on synthetic datasets with induced conflicts—could provide a more comprehensive validation of the method for the intended application.

**Other Comments Or Suggestions:**

- Line 355, the right column, ‘our methods’
- Lin 318, the left column, ‘since the’

**Other Strengths And Weaknesses:**

N/A

**Questions For Authors:**

Please see the above questions. If the authors can adequately address these concerns, I would be happy to reconsider and improve my evaluation of the paper.

**Relation To Broader Scientific Literature:**

The method extends evidential deep learning techniques by incorporating trust discounting to adjust the influence of unreliable views. It also advances previous multi-view classification frameworks such as Trusted Multi-View Classification (TMC) (Han et al., 2021) and its variants (e.g., ETMC, ECML, TMNR, CCML) by explicitly modeling and mitigating the impact of conflicts among views.

**Theoretical Claims:**

I examined the proofs provided for several theoretical claims, particularly focusing on Proposition 3.5 and Proposition 3.6 (detailed in Appendix B.4). And the derivations are consistent with the framework of subjective logic and evidential deep learning, and I did not identify any glaring mathematical errors.

---

> ### Author Rebuttal · Authors · 2025-03-27
>
> We would like to thank you for your valuable feedback,
> and we appreciate your recognition of our novel idea, the effectiveness of the proposed trust discounting method and stage-wise training algorithm, and its adaptivity to the existing evidential framework.
> We have carefully considered each of the concerns raised and have provided detailed responses below.
>
> **Experimental Designs Or Analyses:**
>
> (1) Our work primarily focuses on prediction conflicts, for example, the case shown in Fig. 1, where different views' prediction is different from others.
> We agree with you that explicit evaluation on datasets engineered to induce view conflicts is critical. However, existing literature lacks an ideal method for conflict simulation.
> For example, the only existing work, ECML, attempted to address the conflict issue,
> which randomly replaces original features with instances from other classes (e.g., substituting a "cat" with a "dog")
> to simulate the conflict issue.
> However, this approach has limitations:
> i). unrealistic conflicts: random replacements (e.g., swapping "cat" with "airplane") may create semantically implausible disagreements,
> unlike naturally occurring multi-view conflicts, the one we show in Fig.1.
> ii). distribution mismatch: artificially generated conflicts ignore view-specific feature distributions, potentially biasing evaluation results.
>
> Instead, we propose an alternative conflict simulation method (acknowledging its imperfections, we move detailed results to Appendix D.7, Fig. 6–7 due to the space limitation).
> Our approach injects Gaussian noise into a randomly selected half of the views.
> The intuition is that as noise increases, corrupted views become less informative, approaching random guesses, while uncorrupted views retain (ideally) correct predictions, creating a controlled conflict scenario.
> We train models with noise injected data, ensuring the conflicts arise during the learning phase.
> As shown in Fig. 6–7, our methods (ETF/TF) outperform baselines in handling such conflicts.
>
> (2) We agree that direct conflict measurement is important, and we regret that space limitations prevent us from including these results in the main text.
> In Appendix D.5 and Fig. 4, we measure pairwise view (or inter-view) prediction conflicts and also provide a concrete instance-level example of such conflicts in Fig. 5.
> Our methods TF and ETF builds upon TMC and ETMC, and as shown in Fig. 4, both our methods consistently and significantly reduce conflict ratios across all view pairs.
> For example, the prediction conflict ratio between the GIST view and the pseudo view of EMTC is 0.6, and
> our method ETF reduces it to 0.45, a 15\% absolute improvement.
>
> For the instance level case illustrated in Fig. 5, the original ETMC prediction was Label 4 (while the ground truth was Label 3). After applying ETF, the model correctly shifts higher belief mass to the true label (Label 3).
> Notably, this improvement is not limited to the combined view, it also enhances consistency in view-specific predictions (e.g., pseudo-view and View 2), demonstrating stronger agreement among views.
>
> **Essential References Not Discussed:**
>
> We appreciate the reviewer for highlighting the newly accepted ICLR 2025 work, which we recognize as concurrent research to ours and worthy of discussion in our submission. However, after thorough investigation, we were only able to identify one relevant paper titled ``Trusted Multi-View Classification via Evolutionary Multi-View Fusion". While the authors provide a GitHub link, we note the implementation appears unavailable at this time.
>
> Upon careful analysis, we find their work orthogonal to ours: their primary contribution focuses on enhancing pseudo-view quality through evolutionary fusion, whereas we propose a fundamentally different approach for handling prediction conflict via Trust Discounting.
> That said, we acknowledge potential synergies - our ETF method could potentially incorporate their evolutionary architecture by: i) integrating our referral ENN to further refine the pseudo-view generated by their evolutionary fusion,
> and ii) maintaining our core innovation in conflict resolution while benefiting from their improved pseudo-view feature.
> This complementary combination might yield additional improvements, though we leave such exploration for future work given the current unavailability of their implementation.
>
> In the revised manuscript, we will explicitly cite and discuss the recommended paper, highlighting its relevance and clarifying how our work differs from or builds upon it.
>
> We hope our responses effectively address your concerns and appreciate if you can reconsider and improve the evaluation.

---

### Official Review · Reviewer_icFc · 2025-03-11

**Overall Recommendation:** 3

**Summary:**

The paper addresses the issue of conflicting predictions in multi-view classification tasks, where traditional methods often assume views are equally reliable and aligned. It proposes a computational trust-based discounting mechanism within the Evidential Multi-view Classification (MVC) framework. This mechanism employs instance-wise probability-sensitive trust evaluation based on subjective logic, discounting less reliable view predictions before fusion. The method includes a stage-wise training algorithm and demonstrates improved accuracy and consistency across multiple real-world datasets, outperforming existing MVC approaches.

**Claims And Evidence:**

Yes

**Essential References Not Discussed:**

N/A

**Experimental Designs Or Analyses:**

Empirical experiments conducted on six benchmark datasets convincingly demonstrate that the proposed approach outperforms established baselines in terms of accuracy and consistency metrics, underscoring its practical effectiveness and broad applicability.

**Methods And Evaluation Criteria:**

Yes

**Other Comments Or Suggestions:**

N/A

**Other Strengths And Weaknesses:**

# Strengths

* The paper effectively addresses the practical and often overlooked issue of conflicting predictions in multi-view classification, highlighting scenarios where the standard assumption of equally reliable views does not hold, such as autonomous driving and medical diagnostics.

* A notable strength is the introduction of an innovative trust-based discounting mechanism grounded in subjective logic, which quantifies uncertainty and reliability on an instance-wise basis. This approach uniquely adjusts the fusion process by discounting less reliable view predictions, thus enhancing robustness.

* The authors make a methodological advancement by developing a structured stage-wise training strategy, effectively integrating both functional predictions and referral opinions. This training scheme contributes to improved model robustness, particularly in handling conflicts among views.

# Weaknesses

* While the paper introduces an incremental innovation through a trust-based discounting mechanism, it lacks a clear articulation of how this method differs from and outperforms closely related approaches, such as ECML. Clarifying these distinctions would strengthen the paper's theoretical contribution and impact.

* The paper does not clearly explain how the conflict data is constructed for experiments. The dataset used are all normal multi-view data, which seems contradictory to the conflict scenario the paper aims to address.

* The paper could benefit from a more detailed discussion on scenarios where the proposed method may not perform as expected. Understanding the limitations and failure cases can provide more nuanced insights and guide practical implementations.

* The improvement of the proposed method in some experimental results isn't obvious compared with some other state-of-the-art MVC baselines.

**Questions For Authors:**

* What are the definitions of "referral opinion" and "functional opinion"? Is there any specific difference between their meanings? According to my understanding, they are generated by two separate sets of evidence network parameters.

* Are there known limitations or potential failure cases of the framework that were not discussed in the paper? Understanding where the model might not perform as expected could be crucial for practical implementations.

**Relation To Broader Scientific Literature:**

This paper introduces a novel trust-based mechanism for resolving conflicting information across multiple views in classification tasks. The proposed framework aligns closely with broader efforts in uncertainty estimation and trustworthy multi-view learning, potentially influencing further studies in multi-modal fusion, robustness in decision-making models, and uncertainty-aware ML systems.

**Theoretical Claims:**

Yes

---

> ### Author Rebuttal · Authors · 2025-03-27
>
> We sincerely appreciate the reviewer's valuable feedback and their recognition of:
> our research motivation,
> the novelty of proposed trust-based discounting mechanism and effectiveness of training algorithm.
> We have carefully addressed each of the concerns raised with detailed responses below, and we hope these clarifications satisfactorily answer all questions.
>
> **Questions For Authors:**
>
> (1) We adopt the terminology from the Subjective Logic book,
> distinguishing between referral opinions and functional opinions generated by different types of Evidential Neural Networks (ENNs).
> Specifically, i) Referral opinions serve as reliability assessments, indicating whether a prediction should be trusted,
> ii) Functional opinions directly support decision-making based on their own evidence.
> We confirm your observation that these are generated by two distinct sets of evidence network parameters,
> and there are still some key differences worthy of highlighting.
>
> - As noted in line 266 and line 270, the referral ENN and functional ENN process different input features, referral ENN takes one more input which is the belief masses of functional opinion;
> - Their roles are also different, as in Eq. 5 (Trust Discounting Mechanism) – referral opinions modulate trust, while functional opinions provide evidence for decision making;
> - Referral opinions are always derived from beta distributions (Fig. 2), quantifying the reliability of their associated functional opinions. However, functional opinions may follow either: 1) Dirichlet distributions for multiclass classification or 2) Beta distributions for binary classification.
>
>
> (2) We appreciate this important question. Let us analyze potential limitations from both global and local perspectives.
>
> - From a global perspective, the evidential multi-view classification framework relies on late fusion. This means that there is no early interaction between views during feature extraction. Although effective in many cases, this design might limit knowledge integration when different views provide complementary but partial information about the complete pattern.
> - From a local perspective, our Trust Discounting (TD) module has a potential limitation: it may still use referral opinions with high uncertainty. Since high uncertainty suggests lower reliability, this indicates that our current trust adjustment mechanism could benefit from more fine-grained handling of such cases. We acknowledge this as an area for future improvement.
>
> **Weaknesses:**
>
> (1) The difference between ECML and our work can be summarized as follows,
> - Different Conflict Resolving Mechanism: our method uses a trust discounting module to modulate the trust on the functional opinions, while ECML uses a loss function to harmonize different views' functional opinions (this is already mentioned in the related work section).
> - Our method is built upon TMC and ETMC, so using the same belief fusion method, which is the Dempher-Shafer rule for combining different views from opinion perspective. However, ECML uses a different, evidence averaging based method to fused opinions.
> - To keep the effectiveness of the introduced TD module, we proposed stage-wise training algorithm, which is also different from ECML.
>
> (2) Thank you for raising this important point. A similar concern regarding Experimental Designs or Analyses has been addressed in detail in our response to Reviewer ngw9 (please refer to our first response statement to Reviewer ngw9).
>
> (3) Please refer to our reply to your same question above
>
> (4) Thank you for raising this important point.
> Our method consistently outperforms existing baselines across all six evaluated datasets. We acknowledge that in some cases, the improvements might appear marginal, which we attribute primarily to the following reasons:
> - Limited improvement space: For datasets such as Handwritten, all compared methods already achieve accuracy levels above 99\%, leaving minimal room for further substantial gains.
> - Intrinsic difficulty of certain datasets: For challenging datasets like Caltech, existing state-of-the-art methods typically achieve around 94\% accuracy. Despite this intrinsic difficulty, our method successfully pushes performance beyond this threshold into the 95\% range, demonstrating effectiveness even under constrained conditions.
> - Baseline instability and fluctuations: On datasets such as CUB, baseline methods exhibit performance instability, fluctuating notably between 90\% and 93\% accuracy. In contrast, our approach demonstrates clear superiority by consistently achieving around 94\% accuracy, indicating greater robustness and stability.
>
> Additionally, it's important to highlight that on datasets like Scene, our method achieves a substantial improvement of roughly 4-5\%, while even in the least improved scenario (PIE and HMDB dataset), we still observe a meaningful increase of above 1%.

---

> > ### Comment · Reviewer_icFc · 2025-04-03
> >
> > I acknowledge that I have read both the rebuttal and the reviews from other members of the Reviewers.
> >
> > 1. Some definitions, such as "referral opinion" and "functional opinion," need further clarification. I recommend that you provide clearer definitions for these terms in the revised version to help readers better understand your arguments.
> > 2. Additionally, I recommend incorporating a dataset or toy examples that more clearly demonstrate the advantages of your proposed method. Including such a dataset would strengthen the paper by providing more compelling evidence of the method's effectiveness.
> >
> > Overall, the authors have addressed some of my concerns, and I would like to keep my rating.

---

> > > ### Author Response · Authors · 2025-04-04
> > >
> > > Thank you for your follow-up and constructive feedback.
> > >
> > > 1. We apologize for unintentionally overlooking your question regarding the clear definitions of functional and referral opinions. We appreciate your observation and will include the following formal definitions in the revised manuscript:
> > >     &nbsp;
> > >
> > >     A **Functional Opinion** expresses belief in a model’s own ability to perform a certain task—such as a classification task. It reflects direct trust in the model’s prediction. Let model $A$ be evaluated for its ability to perform a function $f$ (e.g., classification). Then, a functional opinion is a subjective opinion represented as:
> > >     $$\acute{\omega} = [\acute{\mathbf{b}}, \acute{u}, \acute{\mathbf{a}}]$$
> > >
> > >    A **Referral Opinion**, in contrast, expresses belief in a model’s ability to provide reliable referrals regarding another model’s ability to perform a task. It reflects trust in the model’s judgment, not in its own functional capability. Let model $B$ be asked to refer another model $A$ for the function $f$. A referral opinion captures our belief that model $B$ is reliable in making referrals about anther's (i.e., model $A$'s) ability to perform $f$, and is denoted as:
> > >    $$\ddot{\omega} = [\ddot{\mathbf{b}}, \ddot{u}, \ddot{\mathbf{a}}]$$
> > >
> > >     Regardless of whether the opinion is functional or referral, $\mathbf{b}$ is the belief mass vector, $\ddot{u}$ is uncertainty score, with $\mathbf{a}$ being the base rate (i.e., a prior probability distribution over classes, generally a discrete uniform distribution), as already defined in Lines 154–159 in original manuscript.
> > >
> > > &nbsp;
> > >
> > > 2. We appreciate your suggestion to include a dataset or toy example that more clearly demonstrates the advantages of our proposed method.
> > >
> > >     &nbsp;
> > >
> > >     In Section 3, we have already included a toy example to illustrate the behavior and motivation of our method in a controlled setting. Additionally, our method is evaluated on six benchmark datasets, following previous works in Evidential MVC. To further support our claims, we have incorporated a conflict simulation study in the Appendix to analyze the behavior of our method under varying levels of inter-view conflict.
> > >
> > >    &nbsp;
> > >
> > >    Finally, to validate the scalability and real-world applicability of our approach, we also conduct End-to-End training on the large-scale UMPC-Food101 dataset, which consists of 101 classes.
> > >
> > >    &nbsp;
> > >
> > >    We believe this combination of toy example, diverse benchmarks, controlled conflict simulation, and large-scale evaluation provides a comprehensive demonstration of our method’s effectiveness.

---

### Official Review · Reviewer_8ky8 · 2025-03-13

**Overall Recommendation:** 4

**Summary:**

This paper focuses on conflicting multi-view tasks and identifies that misleading predictions with high confidence (low uncertainty) from specific perspectives are key factors in the errors of conflicting multi-view decision-making. To address this issue, the authors propose a view fusion method based on computational trust. This method draws on the principle of trust discounting from Subjective Logic, assigns a binary opinion to each perspective to obtain the trust level for that perspective, and then uses the trust level to weight and fuse the opinions from all perspectives. Extensive experimental results on datasets show that TF has promising results.

**Claims And Evidence:**

Yes

**Essential References Not Discussed:**

No

**Experimental Designs Or Analyses:**

These experiments used six datasets, but only utilized normal datasets, lacking a comparison with the results on datasets with added conflicts.

**Methods And Evaluation Criteria:**

Yes

**Other Comments Or Suggestions:**

We noticed that your main innovation lies in the trust discount fusion from Subjective Logic, which is a reasonable theoretical choice. However, to more clearly demonstrate the unique value of your research, we hope you can further elaborate on the differences between your innovation and the content of the book.

**Other Strengths And Weaknesses:**

Paper strength:
1.The manuscript is well-organized and clearly written, making complex concepts accessible to a broad audience. The visual representations are well-designed and effectively illustrate the proposed method's motivation, methodological designs, and effectiveness of the proposed method.
2.A deep understanding of TF Enhanced Evidential MVC is demonstrated through a comprehensive literature review. The fusion formula of TF(TD + BCF) is derived in detail, which proves that the TF method maximizes the belief quality of the ground truth label, and that the TF fusion gains a larger u than the non-discount fusion.
3.The proposed method is reasonable and innovative. It uses the Trust Discount (TD) in Subjective Logic to assign a binomial opinion to each perspective to measure the trust level, guides the fusion process to generate reliable fused opinions, and solves the wrong predictions caused by opinion conflicts from different perspectives.
4.Extensive experiments on multiple benchmark datasets show significant improvements of the proposed TF method. The effectiveness of the TD module is validated through ablation studies, the capability of handling view conflicts is verified by evaluating multiple models with MVAGT, and the CR demonstrates the TF method's effectiveness in reducing view prediction conflicts.
Paper weakness:
1.Some experimental results need to be added. The paper does not seem to compare the performance after adding conflicts to the datasets. The authors should compare the performance changes of each method on normal datasets and datasets with added conflicts.
2.The detailed description of the network structure is lacking, which is important for reproducing and thoroughly understanding the paper.
3.The paper could be strengthened by providing a more detailed analysis of the limitations of the proposed approach. This factor could be significant for guiding future research and practical applications of the methodology.

**Questions For Authors:**

This paper introduces the Trust Fusion Enhanced Evidential MVC method based on the Trust Discount in Subjective Logic, and conducts extensive evaluations. Regarding the implementation of the Referral Network, in the context of conflicting multi-view learning, "conflict" occurs at random view of each sample, rather than referring to an overall conflict trend in the entire dataset (for example, most samples have a high probability of conflict between view v1 and view v2). After the Referral Network is trained using the training set, when it receives a test sample, it assigns a Binomial opinion to each view of the test sample, thereby assigning a view weight p. The question is whether this weight reflects the overall conflict trend learned from the training set, rather than directly addressing the random inter-views conflict of the current test sample.

**Relation To Broader Scientific Literature:**

This paper introduces a multi-view fusion method based on computational trust. The proposed trust discounting framework is highly consistent with trustworthy multi-view learning and will play a promoting role in the research on multi-view decision-level fusion, reliable uncertainty estimation, and conflicting multi-view tasks.

**Theoretical Claims:**

This paper does not present a theoretical claim or proof. However, through well-designed experiments, it effectively demonstrates the superior performance of the proposed method in addressing conflicting multi-perspective issues compared to existing approaches.

---

> ### Author Rebuttal · Authors · 2025-03-27
>
> We sincerely appreciate the reviewer's recognition of our work's strengths:
> 1) the clear organization and presentation of our manuscript, 2) the comprehensive literature review demonstrating deep domain understanding 3) the methodological innovation in our trust-based fusion approach, and 4) the extensive experimental validation demonstrating effectiveness.
> We have carefully addressed each of the concerns raised and provide detailed responses below. We hope these clarifications fully resolve any remaining questions about our work.
>
> **Weaknesses:**
>
> 1) Thank you for raising this important point. A similar concern regarding Experimental Designs or Analyses has been addressed in detail in our response to Reviewer ngw9 (please refer to our first response statement to Reviewer ngw9).
>
> 2) Regarding network architecture, we maintain consistency with established approaches (TMC, ETMC, ECML, etc.) for fair comparison:
>     - For the six vector-based datasets, we use identical architectures: a single-layer network with Softplus activation to generate functional opinions.
>     - For the Food101 dataset, we employ standard pre-trained models (ResNet50 for images, BERT-base-uncased for text) as feature encoders (as noted in Section 4.4), and referral and functional networks share these encoders on each view. However, each ENN maintains its own output layer (linear + Softplus) to ensure specialized processing while leveraging common feature representations.
>
> 3) Please refer to our response to Reviewer icFc, we will include the limitation our work in the revision.
>
> **Other Comments Or Suggestions:**
>
> We thank you for pointing this out. We believe that clarifying our innovation will make it clearer.
> 1) While the original subjective logic framework proposes a global trust-discounting mechanism, we extend it to operate in an instance-wise manner. This critical advancement enables adaptive handling of varying conflict patterns across different samples, significantly enhancing the framework's flexibility.
> 2) We introduce a novel stage-wise training approach that is essential for keeping the effectiveness of our Trust-Discounting (TD) module. Based on our study, simply incorporating the TD module into existing training frameworks (e.g., ETMC's approach) would actually degrade performance. Our designed training strategy ensures stable optimization and reliable conflict handling.
>
> **Questions:**
>
> We acknowledge that the learned weight p reflects the inherent patterns in the training data. Specifically, when certain views exhibit consistent conflicts, the framework will assign them lower weights.
> However, our design is fundamentally instance-wise, meaning it remains effective even when conflicts lack clear trends across the dataset. For example:
> 1) in our controlled experiments (Appendix D.7, Fig. 6–7), we artificially introduced random conflicts by corrupting 50\% of views with noise—a scenario without view-specific bias.
> 2) under these conditions, such as our method (ETF) still outperformed baselines like ETMC, demonstrating its robustness to sporadic conflicts.
> This validates that the framework adapts not only to systematic conflicts but also to instance-specific conflicts. We appreciate the opportunity to clarify this point.

---

### Official Review · Reviewer_KHzU · 2025-03-14

**Overall Recommendation:** 3

**Summary:**

The paper introduces a novel, trust-based discounting method to enhance the existing Evidential Multi-view Framework in the real-world scenario when the various views are not fully aligned on the labels of some of the examples. The authors leverage a belief-fusion process that considers the reliability of the predictions made by individual views via an instance-wise probability-sensitive trust discounting mechanism. The paper also introduces Multi-View Agreement with Ground Truth, which is a novel metric for measuring the reliability of the predictions.

**Claims And Evidence:**

The Section 3 of the paper is poorly written and organized, which makes it extremely hard to follow (thus, difficult to evaluate the paper's claims). One way to improve its structure would be to:
- simplify sub-section 3.1 to the bare minimum of terminology and notation
- add to sub-section 3.1 the definition of referral and functional opinions, which are used in 3.2 (lines 2013-214) without being introduced (easy to prove by performing a simple search in the doc)
- turn sub-section 3.2 into an illustrative running example; basically show the complete flow of computations (i.e., for Tables 1, 2, and 3, make it clear which values are given inputs and which ones are computed; for this second category, show exactly how they are computed) not only for this particular example, but for all four possible cases of true positive (i.e., the views agree on its label, and the fusion consolidates that decision), a "false positive" (i.e., the current example, for which the naive fusion method decides that it is safe, when -in fact- it is not), a false negative (the opposite of the current example in Table 1, when the naive method wrongly predicts "unsafe"), and a true negative. Having all flows and computation for these four scenarios side by side will allow any reader to truly and understand your approach.
- simplify sub-section 3.3 to a version of Algorithm 1 that can be used to trace what happens with each of the four illustrative example in the new sub-section 3.2

**Essential References Not Discussed:**

N/A

**Experimental Designs Or Analyses:**

See comments from "Methods And Evaluation Criteria" above

**Methods And Evaluation Criteria:**

Please re-organize Section 4 and the appendices in such a way that the new Section 4 matches the very intuitive presentation of Table 1 in [Han et all, 2022]. In the current form of the paper, you split the [Han et al, 2022] table, which, helpfully, includes both accuracy and AUROC, in Table 4 from Section 4 and Table 13 in the APPENDIX.

Very confusingly, your Table 13 in appendix D.2 has significantly lower AUROC values for ETMC. The values in Table 1 of [Han et al, 2022] are much higher (actually quite competitive with yours: on the 6 evaluation domains, the Han paper has AUROCs superior (at times far superior) to TF's and ETF's:
- ETMC:  99.95%,   99.89%,  99.77%,  96.17%,  95.58%, and  99.13%
- TF:        99.32%,  88.99%,  95.90%,   64.56%, 83.59%,  and 53.52%
- ETF:      99.90%,  88.70%,  92.47%,  70,44%,  86.23%,  and  64.41%
These results should be discussed and fully explained in the main paper

**Other Comments Or Suggestions:**

- lines 60-63, left coliumn: please add a reference for the Evidentiary Multi-view framework
- line 57, right column: the "can" in "our method can also enhance ..." makes the statement weak; to strengthen the 3rd claim, you may want to replace it by "does"
- line 190: please avoid unnecessary negatives, such as "is expected to be NOT lower" --> "is expected to be higher"

**Other Strengths And Weaknesses:**

Please explain the discrepancy between (1) ETF's clear superiority wrt ETMC in Table 5 (Fleiss' Kappa), and (2) a version of your Table 13 that is aligned/reconciled with the results in Table 1 from [Han et al, 2022]. If ETMC truly outperforms your approaches on AUROC, wouldn't this also raise concerns about the usefulness of Fleiss' Kappa in this context.

**Questions For Authors:**

- in spite of your claim in line 190 wrt Table 1 (to paraphrase - "the fused opinion is expected to be higher that those of each individual views"), in the final results in Table 3 we still have 0.08 < 0.10 < 0.42 < 0.76. Why is this the case?

**Relation To Broader Scientific Literature:**

The authors seemed to have done an adequate coverage on the broader literature. However, the discrepancy between the results reported in [Han et al, 2022] and those in appendix D.2 are a source of concern.

**Theoretical Claims:**

As all proofs are in the appendices, I did not check them

---

> ### Author Rebuttal · Authors · 2025-03-27
>
> We thank you for your feedback and below is our response to your questions,
>
> 1) Regarding AUROC usage:
> We clarify that our use of AUROC differs fundamentally from TMC [Han et al., 2022].
> While TMC employed AUROC to measure label prediction accuracy,
> we follow the established uncertainty evaluation framework of [Filos et al., 2019],
> where AUROC assesses uncertainty calibration - specifically,
> whether the model's confidence in its predictions correlates with their correctness.
> In other words, it measures the rate of referring the least confident predictions
> to human experts.
>
>     This distinction is important because:
>     - As noted in TMC's results (Table 1), AUROC for label prediction consistently shows high scores ($>$95\%) even when accuracy is modest (67.74\% on Scene15, 65.26\% on HMDB), suggesting limited discriminative value
>     - Subsequent works (ECML, CCML, TMNR) have consequently abandoned this usage
>     - Our application evaluates a fundamentally different capability: the model's ability to identify unreliable predictions through uncertainty scoring.
>
> 2) Writing logic:
> We appreciate the suggestions regarding paper organization. While we will carefully consider these recommendations for future work, we believe the current structure most effectively presents our technical contributions and results.
>
> 3) Wording improvements:
> We thank the reviewer for their thoughtful suggestions on phrasing and will incorporate appropriate refinements while maintaining the paper's technical precision.
>
> 4) Regarding the question:
> We appreciate your close examination of our results. There appears to be a misunderstanding regarding the relationship between Tables 1 and 3, which we would like to clarify:
>    - Table 1 demonstrates the limitation of common belief fusion methods (like BCF [Han, 2022]) where, as we stated in Line 190, "the fused opinion is expected to be higher than those of each individual views" - this represents the problem scenario we aim to solve
>    - Table 3 presents results after applying our Trust Discounting mechanism, which addresses the conflict issue. The values shown (0.08 < 0.10 < 0.42 < 0.76) are:
>         a) expected outcomes of our method,
>         b) demonstrate proper handling of conflicts, and
>         c) represent an improvement over the Table 1.

---

> > ### Comment · Reviewer_KHzU · 2025-04-03
> >
> > Thank you for your detailed answers. I have one follow-up question wrt your comment #4 above:
> > - in the original doc, line 190 says “the uncertainty is expected to be not lower than that of all views to reflect the struggle among different opinions in the presence of conflict.”
> > - at the risk of sounding pedantic, to remove any ambiguity, I would rephrase it as “the FUSED uncertainty is expected to be HIGHER than that of EACH INDIVIDUAL view - to reflect the DISAGREEMENT OF THE CONFLICTING VIEWS.” Is this what you meant?
> > - after applying the ToDs from Table 2 to the Beliefs in Table 1, we get the Beliefs in Table 3. In spite of the discounting, we still have the same conflicting views, right? I.e., Captain & PolarBear still disagree w the Dolphin (but less so, and, in the fused version, the Dolphin’s view now prevails)
> > - according to line 190, I expected that in Table 3 the Fused uncertainty will be higher than the ones of each individual view, which is not the case (hence my confusion)
> >
> > Could you please help me understand what am I missing here? Thank you!

---

> > > ### Author Response · Authors · 2025-04-04
> > >
> > > We thank you for your follow-up and for clarifying the logic behind your question.
> > >
> > > To address your question, we would first like to clarify a preliminary point:
> > > In ETMC [Han, 2022], it was proven that BCF inherently exhibits the characteristic of **always generating lower uncertainty**. As explained in the Subjective Logic book (section 12.2), this is achieved by **ignoring conflicts** and leaning toward a more certain (i.e., lower uncertainty) final prediction.
> > >
> > > Therefore, based on the theoretical proof by Han [2022] and the explanation from the Subjective Logic book, we understand that regardless of the opinions from each view, the fused opinion generated by BCF will always have lower uncertainty than each individual view. In this regard, the results shown in Table 1 and Table 3 are consistent with the expected behavior of BCF.
> > >
> > > However, our method is built upon BCF, so it does not break the theoretical property of always producing a fused opinion with lower uncertainty.
> > > Nonetheless, even under this constraint, after applying the proposed TD, we observe that the uncertainty of the fused opinion increases compared to before — for example, 0.08 in Table 3 vs. 0.01 in Table 1. This supports our Proposition 3.6:
> > >
> > > > "The combined opinion generated by proposed TF (TD+BCF) for conflicting views, will exhibit
> > > greater uncertainty than obtained through fusion with non-discounted functional opinions".
> > >
> > > Proofs for this are provided in the Appendix.
> > >
> > > Additionally, BCF operates in an uncertainty-aware manner, as indicated by Eq.(2) in the original text.
> > > With the TD, the fused opinion changes to the correct one (with more belief mass on “Unsafe”), and its uncertainty also increases (compared to original BCF) — indicating the presence of conflict.
> > > However, the uncertainty remains within a reasonable range (with a magnitude of 0.08), so the opinion can still be considered reliable.
> > >
> > > In summary,
> > > our statement at line 190 reflects an intuitive expectation.
> > > While we acknowledge that “higher than” may appear clearer to some readers, its intended meaning is essentially consistent with the original phrasing “not lower than.” We believe either formulation is acceptable in conveying the intended message.
> > > Meanwhile, without violating the theoretical rule of BCF,
> > > our proposed TD module still adjusts the uncertainty to a more reasonable value, which achieves:
> > > 1) reflecting the presence of conflicts by exhibiting higher uncertainty compared to BCF without TD (proposition 3.6).
> > > 2) providing a reasonable level of uncertainty in the fused opinion, which is slightly higher but still within a reliable range for decision making.

---

### Decision · Program_Chairs · 2025-05-01

**Decision:**

Accept (poster)

**Comment:**

This paper considers the multi-view classification issue under the scenarios where views are of unequal importance and are not fully aligned. The authors design a computational trust-based discounting method to improve current evidential multi-view framework under view conflicts, and utilize an instance-wise probability-sensitive trust discounting mechanism to increase the reliability of predictions made by individual views. Experimental results illustrate the effectiveness.


After rebuttal, three reviewers give 'Weak accept', and one reviewer gives 'Accept'.  Reviewer KHzU holds that current draft is not a nice and clean version. Reviewer icFc hods that the response addresses several of his/her concerns. Combined with the response and reviewers' feedback, I recommend a 'Weak accept'.